# A TOPOLOGICAL VIEW OF RULE LEARNING IN KNOWLEDGE GRAPHS

## ABSTRACT

Inductive relation prediction is an important learning task for knowledge graph completion. One can use the existence of rules, namely a sequence of relations, to predict the relation between two entities. Previous works view rules as paths and primarily focus on the searching of paths between entities. The space of paths is huge, and one has to sacrifice either efficiency or accuracy. In this paper, we consider rules in knowledge graphs as cycles and show that the space of cycles has a unique structure based on the theory of algebraic topology. By exploring the linear structure of the cycle space, we can improve the searching efficiency of rules. We propose to collect cycle bases that span the space of cycles. We build a novel GNN framework on the collected cycles to learn the representations of cycles, and to predict the existence/non-existence of a relation. Our method achieves state-of-the-art performance on benchmarks.

## 1 INTRODUCTION

Knowledge graphs (KGs) are graph-structured knowledge bases that integrate human knowledge through relational triplets. In a KG, nodes represent entities and edges represent relational triplets connecting them. A relational triplet is defined as $(e_h, r, e_t)$, where $e_h$ and $e_t$ are the head and tail entities respectively, and $r$ is the relation between them. KGs have been used in many problems such as recommendation systems (Wang et al., 2018), question answering (Huang et al., 2019; Zhang et al., 2018), biomedical research (Zhao et al., 2020b; Zhu et al., 2020), and zero-shot learning (Kampffmeyer et al., 2019).

Due to the limitation of human knowledge and data extraction algorithms, we cannot thoroughly define and excavate all the entities and relations in a KG (Chen et al., 2020). The incomplete structures and contents of KGs can significantly benefit from an automatic completion algorithm. Early works (Bordes et al., 2013; Yang et al., 2014; Sun et al., 2019) focus on incorporating the attributes of entities. Recent works develop models that are agnostic of entity attributes. They can handle new entities and dynamic KGs, which are quite common.

These entity-agnostic works (Yang et al., 2017; Sadeghian et al., 2019; Teru et al., 2020) are called *inductive relation prediction* models. They predict missing triplets by learning logical rules in KGs. For example, from the KG shown in Figure 1(a), we can learn the rule:

$$\exists X, (X, \textit{part\_of}, Y) \wedge (X, \textit{lives\_in}, Z) \rightarrow (Y, \textit{located\_in}, Z). \tag{1}$$

Based on the learned rule, in Figure 1(b), we can induce the missing triplet $(ManchesterUnited, \textit{located\_in}, Manchester)$ due to the existence of the two-hop path $(Cristiano, \textit{lives\_in}, Manchester)$ and $(Cristiano, \textit{part\_of}, ManchesterUnited)$.[1]

Existing inductive relation prediction works (Yang et al., 2017; Sadeghian et al., 2019) mainly view rules as paths, i.e., sequences of relations connecting two entities of interest. For example, to express the rule shown in Equation 1, these methods need to enumerate all possible 2-hop paths. As Yang et al. (2017) pointed out, the number of learnable parameters of these methods is $O(|R|^T)$, where $|R|$ denotes the number of relation types, and $T$ denotes the maximum length of rules. To avoid

---

[1]Technically speaking, these methods can only learn the "and" operation between relations. We will be interested in expanding to more sophisticated rules, but this is beyond the scope of this paper.

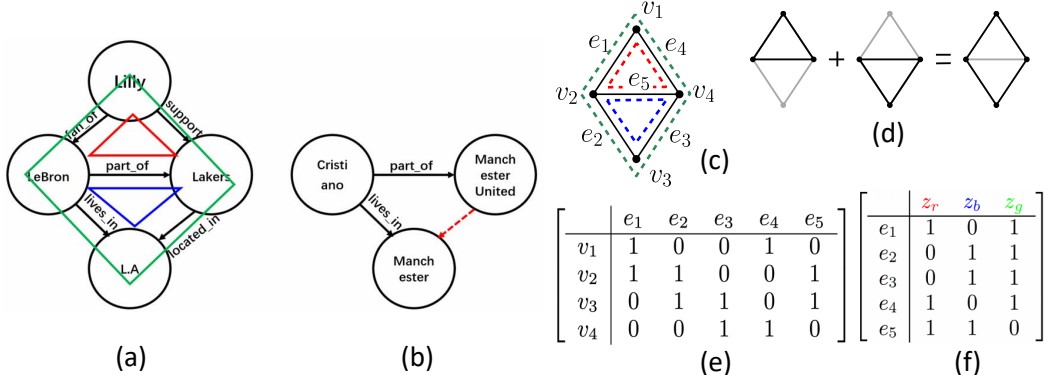

Figure 1: **(a)** and **(b):** examples of inductive relation prediction. **(c)-(f):** illustrations for cycle space and cycle basis. **(c):** A sample graph with three nontrivial cycles, $z_r$, $z_b$ and $z_g$ (highlighted with red, blue and green colors). Any two of the three will form a cycle basis. **(d):** Mod-2 addition of cycles. **(e):** the boundary matrix $\partial$ of the graph in (c). Any cycle $z$ satisfies $\partial z = 0$. **(f):** the cycle incidence matrix of the graph in (c). We show all three columns corresponding to all three nontrivial cycles ($z_r$, $z_b$ and $z_g$). In our algorithm, we only pick the columns corresponding to a chosen cycle basis, e.g., the first two columns when the chosen basis is $\{z_r, z_b\}$.

the exponential-size search space of paths, we tackle the problem from a new perspective: to view logical rules as cycles, and then learn the rules in the space of cycles. In fact, any rule can be considered a cycle by including both the relation path and the target relation itself.

The benefit of using cycles is that there is an intrinsic structure in the space of cycles. Based on the theory of algebraic topology (Munkres, 2018), *the space of cycles is a vector space* under certain assumptions. We can exploit the linear structure of cycle space for efficient rule learning. In particular, we focus on a *basis*, i.e., a set of linearly independent cycles that can represent any cycle. Taking Figure 1(a) as an example, if we choose the red cycle and the blue cycle as the cycle basis, then the green cycle can be represented as their sum. Here "sum" means modulo-2 sum (Figure 1(d)), thus the overlapped triplet $(Lebron, part\_of, Lakers)$ is canceled. By focusing on the cycle basis that spans the cycle space, we decrease the parameter space from exponential to linear[2].

However, the choice of a useful cycle basis is not trivial. Not all cycles/rules are "good", or say, providing a valid relation reasoning. For example, the green cycle in Figure 1(a) reveals the rule:

$$\exists X_1, X_2, (X_1, \text{ support }, Y) \land (X_1, \text{ fan\_of }, X_2) \land (X_2, \text{ lives\_in }, Z) \to (Y, \text{ located\_in }, Z). \quad (2)$$

This rule is not considered good. There is no strong reason that a basketball team $(Y)$ should be located in a city $(Z)$ just because there is a fan $(X_1)$ that supports both the team and a player $(X_2)$ living in the city $Z$. The goal of relation learning is to efficiently comb through all cycles/rules and learn the good ones. A desirable cycle basis should facilitate this process. Intuitively, the elements of the basis should be as close to good cycles as possible. Revisiting the example of Figure 1(a), we prefer the basis consisting of the red and blue cycles (both are good rules) and avoid the green cycle.

To find suitable cycle bases for relation learning, we propose an approach inspired by the theory of topological data analysis (Dey et al., 2010; Chen & Freedman, 2010; Busaryev et al., 2012). These works show that cycle bases generated from shortest path trees have favorable theoretical guarantees. In the context of relation learning, we argue that the good cycles are generally short, and can be easily learned using shortest-path-tree-based cycle bases. We also propose to sample multiple shortest path trees and collect a family of cycle bases. This provides sufficient redundancy and coverage to ensure we learn good cycles/rules efficiently, as experiments will show.

To fully exploit the cycle bases to learn all the good rules, we propose *Cycle Basis Graph Neural Network (CBGNN)*. We build a GNN on a new graph whose nodes represent cycles in the chosen bases, and edges represent their interaction. Through the message passing of the GNN, we are running implicit algebraic operations over the space of cycles. Our method will efficiently learn to represent good cycles, and learn to predict relations in KGs. Experiments on popular inductive relation prediction benchmarks show the efficacy of our method.

---

[2]A cycle basis of size $\beta$ spans a cycle space of size $2^\beta$.

In summary, our contribution is three-fold:

1. We propose, for the first time, to investigate the inductive relation prediction problem through a cycle-based perspective.
2. Inspired by the theory of topological data analysis, we propose to exploit the linear structure of the cycle space, and to compute suitable cycle bases that can best express the rules. This empowers us to explore rule space more efficiently than previous path-based approaches.
3. We propose a novel graph neural network, CBGNN. It runs implicit algebraic operations in the cycle space through the message passing of a GNN, and learns the representation of good cycles and good rules. Experiments show that CBGNN achieves state-of-the-art results on various inductive relation prediction benchmarks.

## 2 RELATED WORKS

**Graph Learning with Topology.** Graph structural information have been shown to enhance graph representation learning (Kipf & Welling, 2016; You et al., 2019; Ye et al., 2019). In recent years advanced topological information, i.e., persistent homology (Edelsbrunner et al., 2000; Edelsbrunner & Harer, 2010), have been applied to graph learning problem. Persistent-homology-based features encode multiscale topological information regarding graph structures in view of different filter functions. They provide additional discriminative power for various tasks such as graph classification (Hofer et al., 2020; Carrière et al., 2020; Hofer et al., 2017; Zhao & Wang, 2019), link prediction (Yan et al., 2021; Bhatia et al., 2019), and node classification (Zhao et al., 2020a). From a different perspective, new graph neural networks have been proposed for high-order graphs, treated as simplicial or cell complexes (Bodnar et al., 2021b;a). Unlike these approaches, we focus on a different task, inductive relation prediction. We exploit the space of cycles and its underlying algebraic structure for better graph representation learning. We believe the cycle-centric design of our graph neural network is generic and can extend to many other tasks beyond relation prediction.

Finally, we note that in topological data analysis, *homology localization*, including computing short cycles representatives of a homology class and computing short cycle bases representing the whole homology group, is well studied theoretically (Chambers et al., 2009; Chen & Freedman, 2011; Dey et al., 2011; Busaryev et al., 2012; Dey et al., 2010; Dey & Wang, 2022). In recent years, new questions have been raised regarding finding short representative cycles for classes in persistent homology (Wu et al., 2017; Dey et al., 2020). Cycle information can complement persistent homology information to achieve better learning power.

**Inductive relation prediction methods.** Inductive relation prediction methods can be divided into two categories: path-based methods, and GNN-based methods. Among path-based methods, AMIE (Galárraga et al., 2013) and RuleN (Meilicke et al., 2018) are classic rule learning methods. These two methods prune the process of rule searching based on strong assumptions on the attribute of rules, thus their performances are not satisfying. To deal with the problem, NeuralLP (Yang et al., 2017) and DRUM (Sadeghian et al., 2019) are proposed to learn the rules in a data-driven manner. However, as pointed in (Sadeghian et al., 2019), these works cannot learn all the rules correctly with a small number of learnable parameters.

GNN-based methods such as GraIL (Teru et al., 2020) and CoMPILE (Mai et al., 2021) predict missing triplets with graph neural networks (GNNs). These methods are appealing as GNNs are able to learn better representations and achieve state-of-the-art performance in general. To predict whether a certain triplet exists in the KG, these methods first extract the corresponding vicinity graph of the triplet and then learn the rules through message passing and GNN scoring. Therefore, they can only predict the triplets one by one, with a rather low computational efficiency. In the experiment part, we will empirically show the efficiency of our framework by comparing it with these models.

## 3 CYCLE SPACE, CYCLE BASIS, AND THE PURSUIT OF SUITABLE BASES

In this section, we explain how to find suitable cycle bases that can facilitate the learning of good cycles/rules. We first introduce the background of the cycle space and cycle basis. Next, we explain our choice of suitable cycle bases, which will be the foundation of our model.

By no means our exposition is comprehensive. For more details, we refer the readers to textbooks on algebraic topology and computational topology (Munkres, 2018; Edelsbrunner & Harer, 2010;

Dey & Wang, 2022). We focus on cycles in undirected graphs, while the definitions generalize to higher dimensions, e.g., simplicial complexes. Furthermore, we focus on the algebraic structures over $\mathbb{Z}_2$ field, which has two elements, 0 and 1, under modulo-2 addition and multiplication. Over $\mathbb{Z}_2$ field, the structure of the space of cycles is simpler and more friendly to computation.

For the rest of the paper, regarding the input KG, we will use node, vertex, and entity interchangeably. We will also use edge and triplet interchangeably. Within this section, we temporarily ignore the relation associated with each triplet. We treat the input KG as an undirected graph $G = (V, E)$, where $V$ and $E$ denote the sets of vertices and edges, respectively.

## 3.1 BACKGROUND: CYCLE SPACE AND CYCLE BASIS

For ease of exposition, we assume the input graph $G$ is connected. The definitions can easily extend to a graph with multiple connected components. An *elementary cycle* is a closed loop, i.e., a sequence of edges, $\{(v_0, v_1), (v_1, v_2), \ldots, (v_{n-1}, v_n), (v_n, v_0)\}$, going through distinct vertices except for the first and the last. A *cycle $z$* is the union of a set of elementary cycles.

The set of all cycles constitute a vector space under modulo-2 additions and multiplications. Figure 1(d) illustrates the mod-2 addition of cycles. There is a nice linear algebra interpretation of the space of cycles. Assume a fixed indexing of all edges and all vertices. The $|V| \times |E|$ incidence matrix, $\partial$, also called the *boundary matrix*, encodes the adjacency relationship between edges and vertices. Any set of edges, called a *chain*, corresponds to an $|E|$-dimensional binary vector, $c$. The $i$-th entry of $c$, $c_i$, is 1 if and only if the chain contains the $i$-th edge, $e_i$. The set of all chains form a vector space called the *chain group*. All chains one-to-one correspond to all possible $|E|$-dimensional binary vectors. Multiplying the boundary matrix to a given chain is equivalent to taking the boundary of the chain. Figure 1(c) and (e) show a sample graph and its boundary matrix. A *cycle* is a chain with zero boundary. Formally, the set of all cycles of $G$, denoted as $\mathcal{Z}_G$, is the kernel space of the boundary matrix, $\mathcal{Z}_G = \ker \partial = \{c \mid \partial c = 0\}$. In the example graph in Figure 1(c), there are 3 different nontrivial cycles, highlighted in red, blue, and green[3].

**Cycle basis.** A *cycle basis* is a basis spanning the cycle space $\mathcal{Z}_G$. Formally, a basis, $Z$, is a maximal set of cycles $\{z_1, z_2, \ldots\}$ such that (1) any cycle in $\mathcal{Z}_G$ can be written as the formal sum of cycles in the basis, $\forall z \in \mathcal{Z}_G, \exists \alpha_i \in \{0, 1\}, s.t. \ z = \sum_{z_i \in Z} \alpha_i z_i$ and (2) cycles in $Z$ are linearly independent, $\sum_{z_i \in Z} \alpha_i z_i = 0 \iff \forall z_i \in Z, \alpha_i = 0$. In Figure 1(c), the red and the blue cycles form a cycle basis. We note that the basis is not unique. The red cycle and the green cycle form another cycle basis of the same graph. However, the number of elements in the basis, $|Z|$, is the same. We call it the *Betti number*, denoted as $\beta$. We have $\beta = |E| - |V| + 1$, and the cycle space has size $2^\beta$.

## 3.2 THE PURSUIT OF SUITABLE CYCLE BASES

The central idea of our approach is to find "suitable cycle bases" to represent the cycle space, so that we can efficiently learn the "good cycles" corresponding to "good rules". In this section, we explain how such suitable cycle bases are constructed. In theory, any basis can represent the whole cycle space, and thus can serve the purpose. However, during learning, we need to look for suitable bases that can easily represent good cycles.

Following the Occam's razor principle, we hypothesize that the good cycles tend to be short, and the desired cycle bases should generally contain short cycles. This is also a manifestation of the principle from the theory of topological data analysis: short cycles are better representatives of topology (Dey & Wang, 2022; Dey et al., 2010; Chen & Freedman, 2010). In fact, this is supported by empirical evidence in relation prediction. It was observed that good rules can be learned even when we exclude long rules from search space (Teru et al., 2020). Motivated by these works, we propose to represent good cycles using *shortest path tree (SPT) cycle bases*, i.e., cycle bases constructed based on shortest path trees. They generally contain relatively short cycles, and can be computed efficiently.

Formally, a *shortest path tree (SPT)* is a spanning tree $T_p \subseteq G$ with root $p$, such that for any vertex $q \neq p$, its path to $p$ within $T_p$ is also its shortest distance path to $p$ within $G$. In other words, $T_p$ is a union of shortest paths from all vertices to the root $p$. A shortest path tree defines a unique cycle basis, which we call the *SPT cycle basis*. As shown in Figure 2(a) and (b), given a shortest path tree,

---

[3]Technically, an empty chain (contains no edges) is also a cycle.

$T_p$, each non-tree edge $e \in E \backslash T_p$ forms an elementary cycle with the tree $T_p$. We construct the basis by enumerating through all non-tree edges and collect all the corresponding elementary cycles. We denote this cycle basis $Z(T_p)$. An SPT cycle basis naturally contains short cycles; each cycle is a composition of an edge $(u, v)$ and two shortest paths - the shortest path from $u$ to $p'$ and the shortest path from $v$ to $p'$. Here $p'$ is the lowest common ancestors of $u$ and $v$ within the rooted tree, $T_p$.

Furthermore, in order to effectively express the rule, we propose to collect a family of SPT cycle bases with different tree roots, to ensure redundancy and sufficient coverage. These bases complement each other and achieve the best learning efficiency in finding good cycles. The hope is that any good cycle can be easily learned by at least one of the bases.

Ideally, we can use the whole vertex set $V$ as roots and build the collection of cycle basis $\{Z(T_p) \mid p \in V\}$. This family of bases has been shown to have theoretical benefit (Dey et al., 2010; Chen & Freedman, 2010). In practice, we cannot afford to construct the bases using all vertices as the SPT roots. We propose to sample vertices that are generally far away from each other. We perform spectral clustering on the graph and use centers of the clusters as the sample vertices, $S$. We hypothesize that these SPT cycle bases will cover the whole graph, and their corresponding cycle bases, $\{Z(T_p) \mid p \in S\}$, will satisfy our needs. We call these bases the *SPT cycle bases family*. As validated in the appendix, these SPT cycle bases provide sufficient coverage of the target edges/triplets, with short cycle representations, compared with random cycle bases.

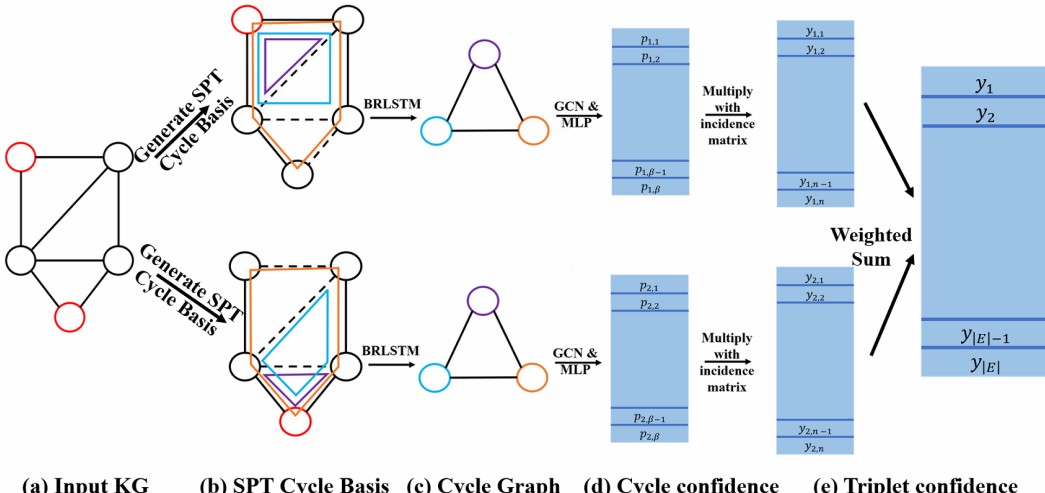

(a) Input KG     (b) SPT Cycle Basis   (c) Cycle Graph   (d) Cycle confidence    (e) Triplet confidence

Figure 2: Architecture of CBGNN. For the input KG shown in (a), we first select the red root nodes using a clustering algorithm and obtain the corresponding SPT cycle bases in (b). Then for every SPT cycle basis. we construct a new graph in (c) where nodes represent cycles in the basis and edges indicate a strong interaction between the two corresponding cycles. The initial node feature vectors are extracted from corresponding cycles using BR-LSTM. With a GCN + MLP, we can learn the confidence value for the nodes (cycles) in (d). And we can compute the confidence value of a given triplet as the max confidence value of cycles passing it. Finally, we use the weighted sum of triplet confidence from different SPT cycle bases as the final triplet confidence.

## 4   CYCLE BASIS GRAPH NEURAL NETWORK (CBGNN)

In this section, we describe how to use the SPT cycle bases family to learn a good cycle representation, to find good rules, and to predict the existence/non-existence of a triplet. We propose a novel graph neural network based on the cycle bases, called the CBGNN. The input of CBGNN is a KG and the target triplets. A target triplet $(e_h, r, e_t)$ refers to a query of whether the relation $r$ exists between entities $e_h$ and $e_t$. A target triplet is labeled positive if it exists in the KG, and negative otherwise. Following the tradition (Teru et al., 2020; Mai et al., 2021), we temporarily add the negative triplets into the input graph. The output of CBGNN is the confidence value of the target triplets. By explicitly constructing a new graph whose nodes correspond to cycles and edges correspond to interaction between cycles, CBGNN learns representations of cycles that best serve the goal of relation prediction.

The overview of our method is shown in Figure 2. Our method has two phases. In the first phase, we construct the cycle bases and build a new graph for each cycle basis (called the *cycle graph*). In the cycle graph, nodes represent cycles in the basis, and nodes are connected if their corresponding cycles have a strong interaction. The information of the cycles can be converted into node features in the new graph for the next phase. Details are provided in Section 4.1.

In phase two (Section 4.2), we build a GNN on the cycle graph to learn the confidence value for cycles. The confidence values for cycles are mapped to the confidence values for target triplets. We construct GNNs for different cycle bases. These GNNs share weights and their aggregation is used to predict the confidence value for the target triplets.

## 4.1 GENERATING THE CYCLE GRAPHS

We construct cycle graphs using the SPT cycle bases family. Recall that in Section 3, we propose to sample vertices at different parts of the input KG and construct SPT cycle bases accordingly. In order to achieve good coverage, these vertices should be selected sufficiently apart from each other. In this way, the family of cycle bases can effectively represent all cycles at different parts of the input KG. In particular, we run spectral clustering on the input graph and partition the nodes into $k$ clusters. Then we take the central nodes of the clusters (the node closest to the cluster center) as the set of sample vertices, $S$.

Using vertices in $S$ as roots, we use the breadth-first-search algorithm to construct $k$ SPTs. The complexity for building each SPT is $O(|V|+|E|)$. [4] For each SPT, $T_p, p \in S$, we construct its cycle basis by going through all non-tree edges. For each non-tree edge, $(u,v) \in E \backslash T_p$, we find the least common ancestor of $u$ and $v$ in $T_p$ in $O(|V|)$ time. In total the complexity for building one cycle basis is $O(|E||V|)$ All the cycles form the desired SPT cycle basis $Z(T_p)$. We now have $k$ cycle bases, each of which has $\beta$ many cycles. The total running time for building $k$ bases is $O(k|V||E|)$.

Note that the input graph may consist of several connected components. The cycle bases of different component graphs are independent of each other. We treat the component graphs as separate input graphs, and generate $k$ SPT cycle bases for each of them. We essentially construct a CBGNN for each component graph, although their weights are all shared.

**Cycle incidence matrix.** We explicitly construct a *cycle incidence matrix* for each SPT cycle basis. This matrix encodes the incidence relationship between cycles and edges in the input KG. It will be used at different stages of our learning; it implicitly encodes interaction between the cycles, and meanwhile, it provides a convenient way to map confidence values between cycles and triplets.

For each of the $k$ constructed cycle bases, we construct the cycle incidence matrix $C_T$ as an $|E| \times \beta$ binary matrix. Each column corresponds to one cycle in the basis. Each row corresponds to an edge/triplet in the input graph. The $(i,j)$-th entry of the matrix is 1 if the $j$-th cycle contains the $i$-th edge, and 0 otherwise. An edge may not be associated with any of the basis cycles and thus has all zeros in the corresponding row. See Figure 1(f) for an illustration. After generating $k$ shortest path trees and their SPT cycle bases, we acquire $k$ cycle incidence matrices: $\{C_T^1, C_T^2, ..., C_T^k\}$.

**Cycle feature.** To use these cycles in learning, we need to extract their attributes/features. We need a feature representation for a cycle based on the relations associated with its triplets. Inspired by existing methods on rule learning (Marcheggiani & Titov, 2017; Vashishth et al., 2019), we propose a recurrent model, *Bi-Relational LSTM (BR-LSTM)*, as the feature generator. It converts a cycle of triplets into a fixed-length feature vector for the CBGNN to use.

During the construction of the SPT cycle bases, we treat the input KG as an undirected graph. But in the generation of cycle feature, we tackle the input KG as a directed graph with different edges/triplets associated with different relations. We assume that information in an edge/triplet flows along both directions, and encode the cycle in a relation-aware manner. We denote by $(u, r, v) \in E$ a triplet connecting nodes $u$ and $v$ with relation $r$. Here $E$ is the set of all the triplets , we assume that an inverse triplet $(v, r^{-1}, u)$ is also included in the KG. Here $r^{-1}$ is defined as the inverse relation of $r$. Formally, we extend the triplet set of the KG as: $E' = E \cup \{(v, r^{-1}, u)|(u, r, v) \in E\}$. An illustration of the construction can be found in the appendix.

---

[4] Note the breadth-first-search algorithm works only because we assume all edges are weighed one.

For each cycle, we can use LSTM to encode the cycle from both directions using triplets in $E'$. Take Figure 1(a) as an example, for simplicity we substitute the relations *part_of*, *lives_in*, and *located_in* with $r_1$, $r_2$ and $r_3$, respectively. The rule can be represented by $(Lebron, r_1, Lakers) \land (Lebron, r_2, L.A) \rightarrow (Lakers, r_3, L.A)$. In practice, we use the non-tree edge in the cycle as the first triplet in the sequence. Therefore we convert the rule into two sequences with the opposite direction: $(Lakers, r_3, L.A), (L.A, r_2^{-1}, Lebron), (Lebron, r_1, Lakers)$ and $(L.A, r_3^{-1}, Lakers), (Lakers, r_1^{-1}, LeBron), (Lebron, r_2, L.A)$. We denote the two sequence as $s_1$ and $s_2$. To encode the two sequences, we adopt a LSTM for each sequence, to capture the contextual information between relations:

$$w_{next_1}, (h_{s_1}, c_{s_1}) = LSTM(w_{s_1}, (h_1, c_1)); \quad w_{next_2}, (h_{s_2}, c_{s_2}) = LSTM(w_{s_2}, (h_2, c_2)).$$

Here, for any $i = 1, 2$, $w_{s_i}$ denotes the input embedding vector for sequence $s_i$. $h_i$ and $c_i$ are the initial hidden state and cell state for sequence $s_i$, they are initialized as zero. $w_{next_i}$ is the output features from the last layer of the LSTM. It is not needed in our setting. $h_{s_i}$ and $c_{s_i}$ are output hidden state and cell state for the whole sequence $s_i$. We use them as the feature vector for each sequence. The final feature vector for the rule and its corresponding cycle, $z$, is $x_z = (h_{s_1} + h_{s_2}) \bigoplus (c_{s_1} + c_{s_2})$, where $\bigoplus$ represents the concatenation of vectors.

## 4.2 GNN Learning with Cycle Graphs

We propose a GNN to exploit the SPT cycle bases to learn representations of good rules and use the learned rules to predict the confidence value of certain triplets. We first build the cycle graphs for the SPT cycle bases, and then learn the confidence value for cycles and triplets.

**Building cycle graphs.** Recall that in Section 4.1, we obtain $k$ cycle bases and $k$ corresponding $C_T$ matrices. For each cycle basis, we construct a new graph in which nodes represent cycles in the cycle basis and edges indicate that the two corresponding cycles have a strong interaction. To measure the interaction between any two cycles in the basis, we compute their overlapping, i.e., the number of triplets they share. In the new graph, each cycle is connected with its top $m$ overlapping neighbors, i.e., the top $m$ other cycles with the most number of shared triplets. To compute the number of shared triplets between all pairs of cycles in the basis, we simply multiply the cycle incidence matrix and its transpose, $C_T^T \cdot C_T$, and read the entries of the resulting $\beta \times \beta$ matrix. For each cycle $z_i$, we inspect its corresponding column in the matrix, and select the top $m$ rows as the cycles to connect.

**Learning cycle representation and confidence.** To learn the representation and confidence values of the desired rules, we apply a classic $L$-layer graph convolutional network (GCN) (Kipf & Welling, 2016) to the constructed cycle graph. The input is the feature vector of cycles generated by BR-LSTM, and the output is the representation of cycles. The message passing by GCN drives the information flow between different cycles. After an $L$-layer GCN, the embedding of a certain node can be viewed as a combination of node representation from its $L$-hop neighborhood. Take Figure 1 as an example, if we take $\{z_r, z_g\}$ as the cycle basis, then $z_r$ and $z_g$ will be the nodes in the new graph. Because they share triplets $e_1$ and $e_4$, there is an edge between the two nodes in the new graph. Then cycle $z_b$ can be learned by the message passing between $z_r$ and $z_g$.

In the $\ell$-th layer of GCN, we can obtain the embedding matrix $X^\ell = [x_1^\ell, x_2^\ell, ..., x_\beta^\ell]$ where $x_i^\ell \in \mathbf{R}^{d_\ell}$ is the representation of node $i$ in the $\ell$-th layer, $\ell = 0, 1, ..., L$. Here, $X^0$ is the initial cycle features from BR-LSTM, and $X^L$ is the node embedding matrix of the final layer. After the $L$-layer GCN, we adopt a two-layer Multi-Layer Perceptron (MLP) followed by a sigmoid function to learn the confidence value for each cycle in the basis: $P = sigmoid(MLP(H^L))$, where $P = [p_1, p_2, ..., p_\beta] \in R^\beta$, $0 \le p_i \le 1$. See Figure 2(c) and (d).

**Learning triplet confidence.** Finally, we compute the confidence values for the triplets of KG based on the confidence values of cycles learned through GNN. We take the max confidence value of cycles/rules that pass a triplet as the confidence value for the triplet. Recall that for each cycle basis, the cycle incidence matrix $C_T$ stores the incidence relationship between cycles and triplets. The $i$-th row of matrix $C_T$ has 1's corresponding to cycles in the basis that pass triplet $e_i$. For triplet $e_i$, its confidence value is computed as $y_i = max(C_T(i, *) \odot P)$, where $C_T(i, *)$ is the $i$-th row of $C_T$, and $\odot$ denotes the element-wise product between two vectors. We obtain the confidence values for all target triplets: $Y = [y_1, y_2, ..., y_n] \in R^n$, where $n$ is the total number of target triplets, $0 \le y_i \le 1, i = 1, 2, ..., n$.

We aggregate the output of the $k$ GCNs to obtain the final triplet confidence. Each GCN is built on one SPT cycle basis and its corresponding cycle graph. We compute the final confidence value of each triplet using a weighted sum of the triplet confidence from different GCNs. Each GCN has one weight value, and the weights are learned during training. Formally, $Y_{final} = \sum_{i=1}^{k} w_i Y_i / \sum_{i=1}^{k} w_i$. We train CBGNN by minimizing the cross-entropy loss on target triplets.

## 5 EXPERIMENTS

We compare our methods with state-of-the-art (SOTA) inductive relation prediction models on popular benchmark datasets. We also use ablation studies to demonstrate the efficacy of different proposed modules in our method. Further experimental details can be found in the appendix.

**Datasets.** We use SOTA benchmark datasets proposed in (Teru et al., 2020; Mai et al., 2021). For inductive relation prediction, the entities in the training set and the test set should not be overlapped. Therefore the training and test sets are totally disjoint graphs. Details are provided in the appendix. Among these datasets, FB15k-237 has $> 200$ relation types, NELL-995 contains an average of 50 relation types, and WN18RR contains $\approx 10$ relation types.

**Baseline.** We compare with SOTA inductive relation prediction methods including (1) path-based methods: NeuralLP (Yang et al., 2017), RuleN (Meilicke et al., 2018) and DRUM (Sadeghian et al., 2019) and (2) GNN-based methods: GraIL (Teru et al., 2020) and CoMPILE (Mai et al., 2021).

**Evaluation.** Similar to (Teru et al., 2020; Mai et al., 2021), we use area under the precision-recall curve (AUC-PR) and Hits@10 scores as the evaluation metrics. To calculate the AUC-PR score, we sample an equal number of non-existent triplets as the negative samples. To evaluate the Hits@10 score, we rank each positive triplet among 50 randomly sampled negative triplets. We run each experiment five times (with different negative samples) and report the mean results.

**Negative sampling.** Following (Teru et al., 2020; Mai et al., 2021), we sample negative triplets by replacing the head (or tail) of a true triplet with a uniformly random sampled entity.

Table 1: AUC-PR scores of inductive relation prediction, the baseline results are copied from (Teru et al., 2020; Mai et al., 2021).

| | | WN18RR | | | | FB15K-237 | | | | NELL-995 | | |
|---|---|---|---|---|---|---|---|---|---|---|---|---|
| Method | v1 | v2 | v3 | v4 | v1 | v2 | v3 | v4 | v1 | v2 | v3 | v4 |
| NeuralLP | 86.02 | 83.78 | 62.90 | 82.06 | 69.64 | 76.55 | 73.95 | 75.74 | 64.66 | 83.61 | 87.58 | 85.69 |
| DRUM | 86.02 | 84.05 | 63.20 | 82.06 | 69.71 | 76.44 | 74.03 | 76.20 | 59.86 | 83.99 | 87.71 | 85.94 |
| RuleN | 90.26 | 89.01 | 76.46 | 85.75 | 75.24 | 88.70 | 91.24 | 91.79 | 84.99 | 88.40 | 87.20 | 80.52 |
| GraIL | 94.32 | 94.18 | 85.80 | 92.72 | 84.69 | 90.57 | 91.68 | 94.46 | **86.05** | 92.62 | 93.34 | 87.50 |
| CoMPILE | 98.23 | **99.56** | **93.60** | **99.80** | 85.50 | 91.68 | 93.12 | 94.90 | 80.16 | **95.88** | 96.08 | 85.48 |
| CBGNN | **98.63** | 97.62 | 89.76 | 97.80 | **96.34** | **96.53** | **96.38** | **95.23** | 82.79 | 94.78 | **96.29** | **94.02** |

Table 2: Hit@10 scores of inductive relation prediction, the baseline results are copied from (Teru et al., 2020; Mai et al., 2021).

| | | WN18RR | | | | FB15K-237 | | | | NELL-995 | | |
|---|---|---|---|---|---|---|---|---|---|---|---|---|
| Method | v1 | v2 | v3 | v4 | v1 | v2 | v3 | v4 | v1 | v2 | v3 | v4 |
| NeuralLP | 74.37 | 68.93 | 46.18 | 67.13 | 52.92 | 58.94 | 52.90 | 55.88 | 40.78 | 78.73 | 82.71 | 80.58 |
| DRUM | 74.37 | 68.93 | 46.18 | 67.13 | 52.92 | 58.73 | 52.90 | 55.88 | 19.42 | 78.55 | 82.71 | 80.58 |
| RuleN | 80.85 | 78.23 | 53.39 | 71.59 | 49.76 | 77.82 | 87.69 | 85.60 | 53.50 | 81.75 | 77.26 | 61.35 |
| GraIL | 82.45 | 78.68 | 58.43 | 73.41 | 64.15 | 81.80 | 82.83 | 89.29 | 59.50 | 93.25 | 91.41 | 73.19 |
| CoMPILE | 83.60 | 79.82 | 60.69 | 75.49 | 67.64 | 82.98 | 84.67 | 87.44 | 58.38 | 93.87 | 92.77 | 75.19 |
| CBGNN | **98.40** | **96.14** | **62.28** | **96.50** | **97.56** | **96.03** | **94.91** | **94.73** | **84.00** | **94.96** | **95.34** | **92.34** |

**Results and discussion.** Table 1 and Table 2 show the AUC-PR scores and Hits@10 scores respectively. Our method outperforms all SOTA baselines in terms of Hits@10 (Table 2). As for AUC-PR (Table 1), our method outperforms nearly all SOTA baselines on FB15K-237 and NELL-995. On WN18RR, CBGNN is a close second, trailing marginally behind CoMPILE, but outperforming the remaining methods significantly. Note that in terms of the number of relationship types, FB15k-237 ($>200$) and NELL-995 ($\approx 50$) are significantly larger than WN18RR ($\approx 10$). They are considered much more semantically complex. *This demonstrates that our novel cycle-based approach has stronger modeling power for KGs with complex semantics.*

**Computational efficiency.** We compare the computational efficiency of our method and the baselines. For a fair comparison, for all methods, we set the training epochs to 100 without early stopping. We run all methods 5 times and report the average time. As shown in Table 3, our method is significantly faster than existing GNN-based methods. Existing GNN-based methods, although demonstrating strong prediction power, are rather expensive. For each training/testing triplet, a GNN method extracts a subgraph within the vicinity and then applies graph convolution. Repeating over all target triplets is rather expensive in practice. On the contrary, our method construct one unified GNN for all target triplets and learn/predict their confidence values simultaneously.

Table 3: Evaluation of computational efficiency (second).

| Dataset | WN18RR v1 | | | FB15K-237 v1 | | | NELL-995 v1 | | |
|---|---|---|---|---|---|---|---|---|---|
| Phase | Preparation | Training | Inference | Preparation | Training | Inference | Preparation | Training | Inference |
| GraIL | 452.36 | 2230.55 | 1.07 | 704.42 | 9026.21 | 1.67 | 402.86 | 3718.22 | 1.79 |
| CoMPILE | 434.45 | 2388.28 | 1.46 | 706.19 | 3809.56 | 2.41 | 479.21 | 2868.38 | 1.23 |
| CBGNN | 601.96 | 952.55 | 0.52 | 437.13 | 901.27 | 0.75 | 379.29 | 175.19 | 0.14 |

**Ablation studies.** We perform ablation studies to validate the efficacy of different proposed modules in CBGNN. We focus on three perspectives, the choice of cycle basis generation and the cycle feature generation. To justify the usage of SPT cycle bases, we compare with a baseline using randomly generated SPTs to build cycle bases, called *CBGNN-Random*. Both CBGNN and CBGNN-Random generate the same number of trees/cycle bases, $k = 20$. To show that sampling multiple trees/cycle bases is necessary, we also add a baseline with a single SPT cycle basis, called *CBGNN-Single*.

For the generation of feature vectors for cycles in the cycle bases, we compare with two baselines which replace BR-LSTM with a bag-of-words-like (BOW) feature vector and a classic LSTM. These method are named *CBGNN-BOW* and *CBGNN-LSTM*, respectively. The BOW feature generates a histogram of different relationship types within a given cycle. The classic LSTM takes a single direction to traverse through the loop instead of two.

Results of the baselines are compared with the proposed CBGNN in Table 4. In terms of cycle bases generation, our method generally outperforms CBGNN-Random. This demonstrates that in most cases, the center nodes of clusters are spread out and are capable of covering the whole graph. Thus for node selection, a clustering algorithm performs much better than random selection. In addition, our method also outperforms CBGNN-Single, showing the necessity to utilize multiple bases to provide better coverage. In the appendix, we will provide more experiments on the influence of the number of SPT cycle bases $k$ on learning cycle representations.

In terms of cycle feature generation, our method outperforms CBGNN-BOW and CBGNN-LSTM on the majority of datasets. The results elucidate the efficacy of our relation-aware feature generation method, BR-LSTM. We were a bit surprised to find that BOW performs well on FB15k-237 and is slightly better than the proposed BR-LSTM. This may be due to the high semantic complexity of this dataset ($>$200 relationship types). The high number of relationship types makes LSTM and BR-LSTM hard to train, whereas BOW may perform robustly under such circumstances.

Table 4: AUC-PR scores of ablation study.

| | WN18RR | | | | FB15K-237 | | | | NELL-995 | | | |
|---|---|---|---|---|---|---|---|---|---|---|---|---|
| Method | v1 | v2 | v3 | v4 | v1 | v2 | v3 | v4 | v1 | v2 | v3 | v4 |
| CBGNN-MLP | 96.33 | 97.49 | 86.86 | 95.22 | 90.90 | 94.07 | 87.01 | 87.78 | 72.29 | 93.35 | 94.63 | 91.29 |
| CBGNN-Random | 97.13 | 76.64 | 87.30 | 93.47 | 96.23 | 96.07 | 93.27 | 94.49 | **83.69** | 93.73 | 96.20 | 92.94 |
| CBGNN-Single | 58.96 | 58.05 | 55.67 | 61.61 | 81.67 | 84.28 | 81.75 | 79.44 | 72.29 | 83.79 | 90.70 | 80.97 |
| CBGNN-BOW | 97.54 | 96.45 | 86.83 | 97.46 | 96.04 | **97.61** | **96.85** | **97.00** | 75.31 | 90.25 | 91.00 | 87.53 |
| CBGNN-LSTM | 98.26 | 97.04 | 89.69 | 97.75 | 95.86 | 91.46 | 94.56 | 92.47 | 71.85 | 93.33 | 93.74 | 85.78 |
| CBGNN | **98.63** | **97.62** | **89.76** | **97.80** | **96.34** | 96.53 | 96.38 | 95.23 | 82.79 | **94.78** | **96.29** | **94.02** |

# 6  CONCLUSION

We provide a novel GNN-based method for inductive relation prediction in knowledge graphs, and propose a cycle-centric approach that treats rule learning as a cycle learning problem for the first time. We exploit the intrinsic linear structure of the space of cycles and learn suitable cycle bases to represent the rules. The learning of cycle representation is carried out via a GNN that passes messages between cycles instead of nodes. Our approach achieves SOTA performance on various inductive relation prediction benchmarks, and provides a novel perspective in incorporating advanced topological information into graph representation learning. Also, our method can naturally be extended to tasks beyond relation prediction.

**Reproducibility Statement.** The details of CBGNN are mentioned in Section 4. The implementation details are mentioned in Section 5 and Section A.2 of Appendix. The details of the data and the used computation resources are described in Section A.2 of Appendix.

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

## A APPENDIX

### A.1 AN EXAMPLE OF BR-LSTM CONSTRUCTION

In this section, we provide an example of the construction of BR-LSTM proposed in Section 4.1. Recall that in the generation of cycle feature, we tackle the input KG as a directed graph with different edges/triplets associated with different relations. We assume that information in an edge/triplet flows along both directions, and encode the cycle in a relation-aware manner. We denote by $(u, r, v) \in E$ a triplet connecting nodes $u$ and $v$ with relation $r$. Here $E$ is the set of all the triplets, we assume that an inverse triplet $(v, r^{-1}, u)$ is also included in the KG. Here $r^{-1}$ is defined as the inverse relation of $r$. Formally, we extend the triplet set of the KG as: $E' = E \cup \{(v, r^{-1}, u) | (u, r, v) \in E\}$. An illustration is shown in Figure 3. Through the triplets in $E'$, we can convert the rule shown in Figure 1 (b) into two opposite sequences: $(Cristiano, part\_of, ManchesterUnited)$, $(ManchesterUnited, located\_in, Manchester)$, $(Manchester, lives\_in^{-1}, Cristiano)$ and $(ManchesterUnited, part\_of^{-1}, Cristiano)$, $(Cristiano, lives\_in, Manchester)$, $(Manchester, located\_in^{-1}, ManchesterUnited)$.

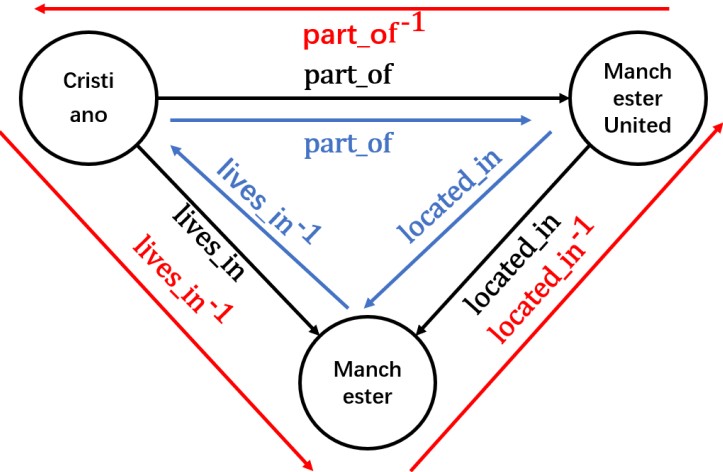

Figure 3: An example of BR-LSTM construction for Figure 1 (b).

## A.2 EXPERIMENTAL DETAILS.

**Datasets.** The datasets used in our settings are subsets of KG WN18RR (Toutanova & Chen, 2015), FB15k-237 (Dettmers et al., 2018), and NELL-995 (Xiong et al., 2017). Teru et al. (2020) generate these datasets by sampling disjoint subgraphs from the original datasets. For inductive relation prediction, the entities in the training set and the test set should not be overlapped. To evaluate the robustness of models, Teru et al. (2020) sample four different pairs of training sets and test sets with the increasing number of nodes and edges. The details of the benchmark datasets are shown in Table 5.

**Experimental details.** We adopt a 2-layer BR-LSTM to generate feature vectors for all the cycles in a cycle basis. Its output feature vector dimension is set to 20. A 2-layer GCN (Kipf & Welling, 2016) is adopted for the message passing of cycle basis, where ReLU serves as the activation function between GCN layers. We combine 20 different shortest path trees to learn the good rules in the given dataset[5]. In the cycle graph, we select the top 2 most related cycles for each cycle. For all the modules, Adam is used as the optimizer, the dropout is set to 0.2, the epoch is set to 100, the learning rate is 0.005 and the weight decay is 5e-5. We follow the settings in (Teru et al., 2020; Mai et al., 2021), that is, to view all the existing triplets in KG as positive triplets and sample negative triplets by replacing the head (or tail) of the triplet with a uniformly sampled random entity. We use binary cross-entropy loss as the loss function with the negative sampling method. Considering that some inductive test sets contain few cycles, which leads to the inconsistent performance between the inductive test sets and original training sets, we use the inductive training set as the validation set (while the training set and the test set are the same with (Teru et al., 2020; Mai et al., 2021)). We

---

[5]In seldom cases such as NELL-995 v2, considering that we can significantly benefit from more shortest path trees, we combine 50 cycle bases for relation prediction.

Table 5: Statistics of inductive benchmarks.

| | | WN18RR | | | FB15K-237 | | | NELL-995 | | |
|---|---|---|---|---|---|---|---|---|---|---|
| | | RELATIONS | NODES | LINKS | RELATIONS | NODES | LINKS | RELATIONS | NODSE | LINKS |
| V1 | TRAIN | 9 | 2746 | 6678 | 183 | 2000 | 5226 | 14 | 10915 | 5540 |
| | TEST | 9 | 922 | 1991 | 146 | 1500 | 2404 | 14 | 225 | 1034 |
| V2 | TRAIN | 10 | 6954 | 18968 | 203 | 3000 | 12085 | 88 | 2564 | 10109 |
| | TEST | 10 | 2923 | 4863 | 176 | 2000 | 5092 | 79 | 4937 | 5521 |
| V3 | TRAIN | 11 | 12078 | 32150 | 218 | 4000 | 22394 | 142 | 4647 | 20117 |
| | TEST | 11 | 5084 | 7470 | 187 | 3000 | 9137 | 122 | 4921 | 9668 |
| V4 | TRAIN | 9 | 3861 | 9842 | 222 | 5000 | 33916 | 77 | 2092 | 9289 |
| | TEST | 9 | 7208 | 15157 | 204 | 3500 | 14554 | 61 | 3294 | 8520 |

run all the baseline methods with a cluster of two Intel Xeon Gold 5128 processors, 192GB RAM, and one GeForce RTX 2080 Ti graphics card.

## A.3 FURTHER EXPERIMENTS

**Experiments with the same settings.** In Section 5, we have compared the performance of our model with the baseline results copied from (Teru et al., 2020; Mai et al., 2021), as shown in Table 1 and Table 2. For a fair comparison, we set Grail and CoMPILE as the same experimental settings as ours, and record the result in Table 6 and Table 7. Similar to the observation in Section 5, CBGNN consistently achieves the state-of-the-art results in the evaluation of Hit@10 scores and outperforms the majority of benchmark datasets when it comes to AUC-PR scores. The results further show the effectiveness of our proposed method.

Table 6: AUC-PR scores of inductive relation prediction, we keep the settings of baseline methods as the same as ours and run these methods five times for the average scores.

| Method | WN18RR | | | | FB15K-237 | | | | NELL-995 | | | |
| | v1 | v2 | v3 | v4 | v1 | v2 | v3 | v4 | v1 | v2 | v3 | v4 |
|---|---|---|---|---|---|---|---|---|---|---|---|---|
| GraIL | 96.09 | 95.92 | 85.86 | 94.02 | 83.98 | 90.66 | 90.17 | 84.74 | 82.34 | 92.35 | 91.45 | 82.88 |
| CoMPILE | 98.56 | **99.98** | **94.04** | **99.85** | 83.45 | 92.17 | 90.91 | 91.39 | 78.07 | 94.07 | 95.69 | 83.40 |
| CBGNN | **98.63** | 97.62 | 89.76 | 97.80 | **96.34** | **96.53** | **96.38** | **95.23** | **82.79** | **94.78** | **96.29** | **94.02** |

Table 7: Hit@10 scores of inductive relation prediction, we keep the validation datasets of baseline methods as the same as ours and run these methods five times for the average scores.

| Method | WN18RR | | | | FB15K-237 | | | | NELL-995 | | | |
| | v1 | v2 | v3 | v4 | v1 | v2 | v3 | v4 | v1 | v2 | v3 | v4 |
|---|---|---|---|---|---|---|---|---|---|---|---|---|
| GraIL | 84.04 | 81.63 | 60.65 | 75.34 | 64.63 | 82.00 | 82.54 | 78.16 | 55.00 | 93.27 | 89.74 | 73.94 |
| CoMPILE | 82.71 | 80.82 | **62.56** | 75.92 | 69.75 | 82.52 | 82.95 | 85.46 | 62.00 | 91.18 | 93.75 | 74.29 |
| CBGNN | **98.40** | **96.14** | 62.28 | **96.50** | **97.56** | **96.03** | **94.91** | **94.73** | **84.00** | **94.96** | **95.34** | **92.34** |

**The influence of $k$.** In this paragraph, we do experiments on the influence of the number of the shortest path trees $k$ which are used to learn the suitable cycle basis. As is shown in Figure 4, CBGNN performs badly with a single cycle basis. However, its performance grows quickly as $k$ increases from 1, and gradually converges after $k$ is large enough (10 for smaller graphs like WN18RR v1 and FB15k-237 v1, and 20 for larger graphs like WN18RR v2 and FB15k-237 v2).. The experiments show that it is crucial to utilize multiple bases to guarantee better coverage. However, after $k$ grows to a certain extent, the root nodes will be well spread out, and contain enough information to cover the whole graph. Therefore, the model hardly benefits from the increase of $k$ after it is larger than a certain threshold. One important factor that may influence the threshold is the size of the input graph. For smaller graphs, we only needs a small number of SPT cycle bases to have a better coverage of the graph. While for larger graphs, we may need more SPT cycle bases. But as shown in Table 1 and 2, 20 SPT cycle bases are enough to gain a state-of-the-art results in most situations.

**Evaluation of shortness.** Recall that in Section 3.2, we follow the Occam's razor principle and hypothesize that the desired cycle bases should generally contain short cycles. In this paragraph, we evaluate the shortness of the SPT cycle bases on various datasets and analyze the correlation between the shortness and performance of different choices of cycle bases. To be specific, we draw histograms to evaluate the minimum length of cycles that pass a triplet. We compare different choices of cycle bases, including a single cycle basis, 10 randomly chosen cycle bases, and 10 cycle bases chosen by the clustering algorithm, which are denoted by "Single", "Random-10", and "Cluster-10" respectively. The histograms are shown in Figure 5. In the histogram, the x-axis denotes the minimum length of cycles that pass a certain triplet, and the y-axis represents the proportion of triplets with a certain minimum length of cycles among all triplets.

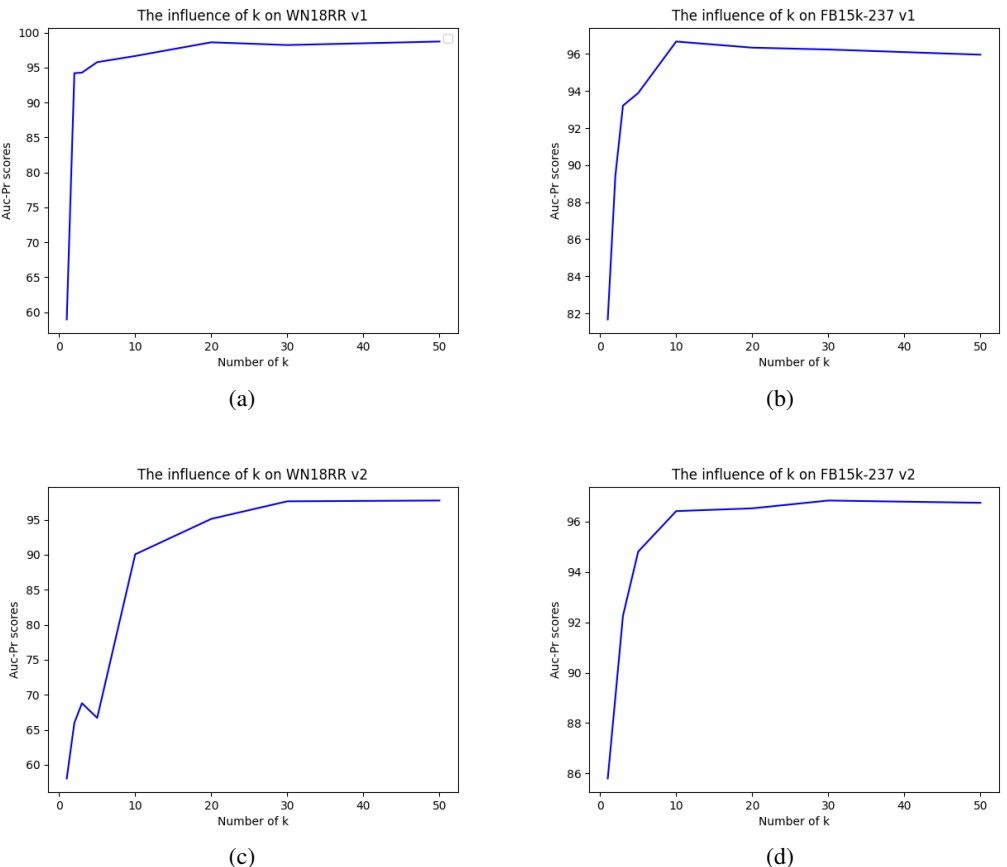

Figure 4: The influence of $k$ on WN18RR v1, v2, and FB15k-237 v1, v2.

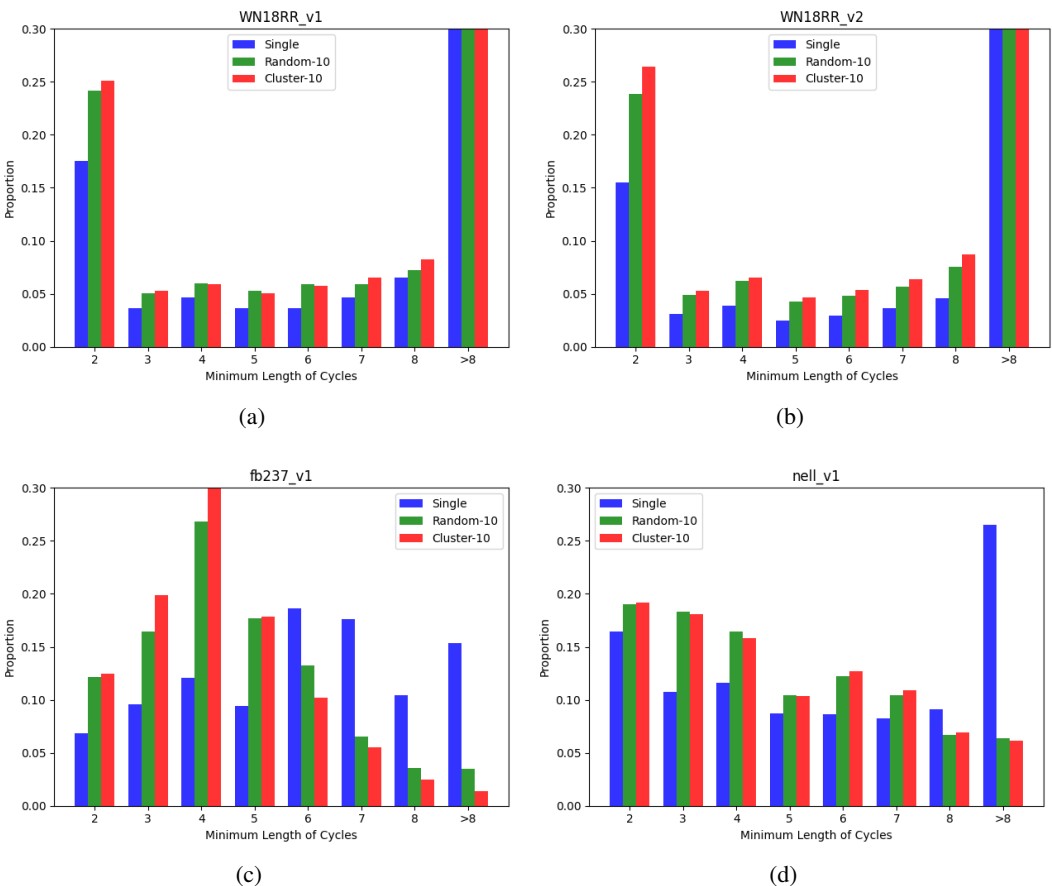

Figure 5: Histogram of shortness on different datasets

As shown in Figure 5, the cycle bases selected by the clustering algorithm generally contain small cycles compared with the randomly selected cycle bases or the single cycle basis. We can find that in Table 4, CBGNN outperforms CBGNN-Random on most datasets. Another interesting observation is that in Figure 5 (d), the randomly selected cycle bases perform comparably with the cycle bases generated using the clustering algorithm in terms of shortness on NELL-995 v1. Recall that in Table 4, the performance of CBGNN-Random slightly beat CBGNN on NELL-995 v1. The above observations show the correlation between the shortness of cycle bases and their performance.

