# OpenReview forum: "A Topological View of Rule Learning in Knowledge Graphs"
_ICLR.cc/2022/Conference — ICLR 2022 Submitted_

### Official Review · Reviewer_pNaT · 2021-10-31

**Correctness:** 4
**Technical Novelty And Significance:** 4
**Empirical Novelty And Significance:** 3
**Recommendation:** 5
**Confidence:** 3

**Main Review:**

strengths:
- The paper is well-written and easy to follow.
- Treating rule learning as a cycle learning problem is very interesting and novel.
- The proposed method is shown to be effective and efficient. In terms of experimetns, the paper includes detailed experimental setup, a comprehensive set of baselines and ablation studies.

weaknesses:
- I can't understand why there is difference between Table 1/2 and Table 6/7. Is this due to random seed? Some of the entries have really big differences. For example, (NELL-995 v1, CoMPILE) in Table 7 and Table 2 are 62.00 and 58.38 respectively. Do we need some statistical testing here or reporting the variance among different runs?
- The paper argues that the proposed method might work better on datasets with complex semantics, i.e. with more complicated relations. Given this, I would suggest running experiments on datasets like https://allenai.org/data/tuple-kb, which contains more than 1000 relations.


**Summary Of The Paper:**

This paper addresses the problem of inductive relation prediction in knowledge graphs. The authors propose a novel GNN-based method. Treating rule learning as a cycle learning problem is very interesting. The experiments on popular knowledge graph completion datasets verify the effectiveness of the proposed method.

**Summary Of The Review:**

The paper proposes to treat rule learning as cycle learning in knowledge graphs. The approach is very interesting and novel with solid experiments showing its effectiveness. I would suggest accepting this paper.

---

> ### Author Response · Authors · 2021-11-23
> **Response to Reviewer pNaT**
>
> > Q1. “I can't understand why there is a difference between Table 1/2 and Table 6/7. Is this due to random seed? Some of the entries have really big differences. For example, (NELL-995 v1, CoMPILE) in Table 7 and Table 2 are 62.00 and 58.38 respectively. Do we need some statistical testing here or reporting the variance among different runs?”
>
>  A1. **Different performance in Table 1, 2, and Table 6,7.**
> The main difference is that we use different validation datasets because some inductive test sets contain few cycles, which leads to inconsistent performance between the inductive test sets and original training sets.
> Therefore, we compare our methods with both the results listed in the original paper (Table 1, 2) and the methods with the same validation datasets (Table 6,7). The different performances in the two tables may originate from this setting. Taking (NELL-995 v1, CoMPILE)  as an example, there are few cycles and few edges in the inductive test sets, therefore, it will be hard for the GNNs to learn, the performance on the test sets may perform consistently with the new validation set (which are from the same entity groups) rather than the original validation set.
>
> > Q2. “The paper argues that the proposed method might work better on datasets with complex semantics, i.e. with more complicated relations. Given this, I would suggest running experiments on datasets like https://allenai.org/data/tuple-kb, which contains more than 1000 relations.”
>
> A2. **Add new dataset.**
> Thanks for your suggestion, considering that the dataset is not originally designed for inductive relation prediction, it will take much time to generate a standard inductive relation prediction benchmark. The evaluation of these datasets can be an interesting future work to explore.

---

> > ### Comment · Reviewer_pNaT · 2021-11-29
> > **Thanks for the response but my concerns remain**
> >
> > Thank you for the response but my concerns still remain. After reading A1, I got a bit confused about the experimental setup. Also, I think datasets other than WN18RR/FB15K-237/NELL are worth trying out and shouldn't take much time. I think the paper can be improved a lot by elaborating on the experimental setup and making the exposition on methodology more clear.

---

### Official Review · Reviewer_weC8 · 2021-11-01

**Correctness:** 2
**Technical Novelty And Significance:** 2
**Empirical Novelty And Significance:** 2
**Recommendation:** 1
**Confidence:** 4

**Main Review:**

The approach put forward in this paper is not convincing since it's convoluted (W1), neither well-argued for and nor convincing (W2), partly unclear (W3), there is no analysis or theoretical insight (W4), and practical benefits and costs ultimately remain unclear (W5).

W1 (convoluted). In a nutshell, (1) run spectral clustering to obtain central nodes, (2) compute shortest-path trees from these nodes, (3) construct a "cycle basis" from each tree, (4) run a BILSTM to encode each cycle in the basis, (5) construct a cycle graph from each cycle basis, (6) run a GNN to predict cycle confidences, (7) repeat (1)-(6) k times and average, (8) predict the cycle confidence of the highest-weighted cycle that contains a triple as the triple confidence, (9) train this using negative sampling.

W2 (neither well-argued for and nor convincing). Many of the steps taken in this approach are heuristic and justified in a rather hand-wavy away; even key points (such as "what is a good cycle" and "why a complete cycle basis") are not precisely described. There is not clear, convincing argument for the approach. In fact, the ultimate approach suggests that if there is a single triple on a cycle that obtains a high-confidence prediction, so do all other triples on the cycle. This point alone makes the entire approach questionable: "son(x,y) and son(y,z) -> related(x, z)" (always true) should have a different confidence than "related(x,z) and son(y, z) -> son(x,y)" (clearly not always true). Another point is that it is known that useful rules may not necessarily be paths; e.g., they may require constants (e.g., "child(x,y) and gender(x,'male')->son(x,y)").

W3 (partly unclear). The paper talks largely about how cycle bases are constructed for a given KG, i.e., for the positive edges. But what happens with the negative edges? They clearly do not appear in any cycle of the cycle base. So how is prediction actually performed +++? For the test set, is leakage of information from the test triples prevented? (+++ needs to be performed for each test triple separately, hiding all other test triples---else information is leaked.)

W4 (no analysis or theoretical insight). Expressivity or properties of the proposed method are neither discussed nor rigidly analyzed.

W5 (practical benefits and costs unclear). First, the method does not fully enclose its cost. What is the time/space requirement for each of the steps 1-9 above? It seems like the method is very costly and does not scale to large graphs. It's unclear why there would be any performance benefit compared to just using a GNN (since it's a part of the proposed method and the graph is larger). Second, it's completely unclear why the proposed method shows strong empirical results; there is no discussion or analysis in the paper. Except there is test leakage (e.g., by constructing a single graph with all test triples; see also +++ in W3), in which case the proposed method is obviously faster and likely to be much better empirically. Apart from the points above, the usefulness of the setting used in the study is mysterious to me (it's taken from prior work): clearly, in the context of the knowledge graphs considered in the experimental, there is entity overlap between train and test (e.g., the higher-level wordnet synsets or the types and locations for Freebase). So why perform reasoning without? The resulting graphs are likely to be quite unnatural/unrealistic.


**Summary Of The Paper:**

The paper proposes an approach to predict the confidence of missing edges in a knowledge graph in the purely inductive setting, in which the test knowledge graph does not share any entities with the training graph. The key idea is to use cycles (or more specifically, cycle bases) as the basis for such reasoning; the paper builds and evaluates a model around this idea.


**Summary Of The Review:**

The approach put forward in this paper is not convincing since it's convoluted (W1), neither well-argued for and nor convincing (W2), partly unclear (W3), there is no analysis or theoretical insight (W4), and practical benefits and costs ultimately remain unclear (W5).

---

> ### Author Response · Authors · 2021-11-23
> **Response to Reviewer weC8**
>
> > Q1 "The approach put forward in this paper is not convincing since it's convoluted"
>
> A1. Please refer to the paragraph regarding **“Q3 The complicatedness and computational efficiency of our model”** in the Overall Response
>
> > Q2. "What is a good cycle?"
>
> A2. Please refer to the paragraph regarding **“Q2 Length of good cycles”** in the Overall Response
>
> > Q3. "Why a complete cycle basis?"
>
> A3. A cycle basis is the maximal set of cycles to represent the complete cycle space without redundancy; any cycle in the cycle space can be uniquely written as the formal sum of cycles in the basis. Therefore, we can only make sure all relevant cycles can be learned by using a complete cycle basis.
>
> > Q4. " "son(x,y) and son(y,z) -> related(x, z)" (always true) should have a different confidence than "related(x,z) and son(y, z) -> son(x,y)" (clearly not always true)."
>
> A4. **The direction/sequence of rules.**
> In our settings, for every cycle/rule, we only use one sequence to represent it, that is, to set the target triplet (the non-tree edge in practice, as stated in Section 4.1) as the start token of BRLSTM. Our aim is to predict the target triplet through the other triplets in the cycle. For example, in Figure 3 in the appendix, if we want to predict the existence of the triplet “(Cristiano,  part of, ManchesterUnited)”, we set it as the first token in the BRLSTM, as explained in Section A.2.
>
> > Q5. “Another point is that it is known that useful rules may not necessarily be paths; e.g., they may require constants (e.g., "child(x,y) and gender(x,'male')->son(x,y)").”
>
> A5. **Setting of rules in inductive relation prediction.**
> We admit that our methods cannot cover all types of rules, however, we are focused on the rules which can be used for inductive learning, i.e. generalizable to unseen entities. That is the same as how previous works about GNN+inductive learning  (Sadeghian et al., 2019; Teru et al., 2020; Mai et al., 2021) did. The given example is clearly an entity-related rule, it depends on the entity “male”, so it does not fit the inductive setting (if we see this entity “male” or “x’ ‘y’， in the training data, we will not test it in the test data).
> However, it does not mean our method cannot capture such rules in any circumstances, because beyond rules, our method is also based on the semantics learned from  GNN and BRLSTM.

---

> > ### Author Response · Authors · 2021-11-23
> > **Additional Response to Reviewer weC8**
> >
> > > Q6. "The paper talks largely about how cycle bases are constructed for a given KG, i.e., for the positive edges. But what happens with the negative edges?"
> >
> > A6. **The setting of negative triplets.**
> > We first construct the SPT with the positive edges and add the negative edges and the non-tree positive edges to the SPT to generate different cycles. This is exactly the same setting with existing works (Teru et al., 2020; Mai et al., 2021). These methods generate the subgraph for every triplet. These subgraphs are constructed by all the positive (existing) triplets (and a temporarily added negative triplet if the subgraph corresponds to the negative triplet). Therefore, in our settings, we generate the SPT with all the positive triplets and treat the negative edges as non-tree edges, which will only occur once in the cycle basis.
> > Actually, it is the same in the path-based methods (Yang et al., 2017; Sadeghian et al., 2019). They need to include the negative triplets in the path-like (chain-like) rule to encode the rule. The process is equal to temporarily adding the negative triplets in the subgraphs.
> >
> > > Q7. "For the test set, is leakage of information from the test triples prevented? "
> >
> > A7. **The setting of test triplets.**
> >
> > Our setting does not have information leakage for the test triplets.
> >
> > **Need to generate the graph with test triplets.** Since the test sets do not overlap with the training graph, in a setting of inductive relation prediction, we need to first build the graph based on all other test triples while leaving the target one out. However, if the target triplet is necessary to build the graph, the common way is to first assume it exists and then test the probability of its existence. (The process is actually consistent with the training process)
> >
> > **No test leakage during and after cycle generation.** Existing works (Teru et al., 2020; Mai et al., 2021) extract the enclosing subgraphs around target triplets with all other positive test triplets and add the target triplet (no matter positive or negative) to the subgraph for testing. Following their settings, for a specific target test triplet, we use some positive test triplets to generate the SPT and add the target triplet as a non-tree edge (no matter positive or negative) to generate the cycle. By combining cycles from all target triplets (positive and negative), we get the SPT cycle bases. In this process, the label of the target triplet is not seen, and its label does not impact the cycle generation. After the generation of cycles, there will be no addictive information of entities or triplets, the only remaining information is the relations in the cycles.
> >
> >
> > > Q8 "there is no analysis or theoretical insight"
> >
> > A8. Our method is inspired by the theory insights of algebraic topology and computational topology (Munkres, 2018; Edelsbrunner & Harer, 2010; Dey & Wang, 2022). Any relevant cycle of the input KG is guaranteed to be represented by cycles in a cycle basis. In other words, we can represent and learn any relevant cycle, and thus any needed rule, with our CBGNN.
> >
> > > Q9. " What is the time/space requirement for each of the steps 1-9 above? It seems like the method is very costly and does not scale to large graphs. "
> >
> > A9. Please refer to the paragraph regarding **“Evaluation of speed / computational efficiency”** in the Overall Response.
> >
> >
> > > Q10. “Apart from the points above, the usefulness of the setting used in the study is mysterious to me (it's taken from prior work): clearly, in the context of the knowledge graphs considered in the experimental, there is entity overlap between train and test (e.g., the higher-level wordnet synsets or the types and locations for Freebase). So why perform reasoning without? The resulting graphs are likely to be quite unnatural/unrealistic.”
> >
> > A10. **Use of inductive relation prediction.**
> > Our work is not the first or the last to do inductive learning. It is a well-known setting in the KGC community. The values of this setting have been demonstrated a lot of times in previous works (Sadeghian et al., 2019; Teru et al., 2020; Mai et al., 2021). So we believe that is not a point that we need to emphasize too much here. In short, there are many practical cases that inductive learning is necessary. For example, in a dynamically growing medical KG, suddenly we get a new drug which never occurred in training data before.

---

> ### Comment · Reviewer_weC8 · 2021-11-25
> **Thoughts on author feedback**
>
> The response does not clear up my concerns. In more detail, my concerns were:
>
> W1 (convoluted). The authors argue that their approach uses prior algorithms in most of each steps. This does not make the approach less convoluted, however.
>
> W2 (neither well-argued for and nor convincing). I am still neither convinced that it's a good idea to use cycles in the first place nor of using a complete cycle basis (instead of well-chosen cycles/paths). My concerns around learning erroneous rules ("related(x,z) and son(y, z) -> son(x,y)") have not been discussed.
>
> W3 (partly unclear). Partly addressed. The authors test the target triple against 50 negatives (by perturbing head/tail). A more convincing approach would be to test it against all entities: this is what would be needed to actually infer new triples in practice and is common in transductive KGC evaluation. (Prior work also used 50 negatives in the setting of this paper, to be fair.) It's also not clear to me how all this relates to the statement "On the contrary, our method construct one unified GNN for all target triplets and learn/predict their confidence values simultaneously." made in the paper. How would one use such an approach in practice?
>
> W4 (no analysis or theoretical insight). Not addressed.
>
> W5 (practical benefits and costs unclear). The authors did not respond to my concerns around the asymptotic space/time complexity or reasons for the good empirical performance. As for inductive reasoning, I questioned whether the setting is useful in practice. The authors do not give a convincing argument, other than that prior work considered it. The example given in the response actually highlights the point made in my review, namely that entity overlap between train and test is likely. Quoting: "[...] in a dynamically growing medical KG, suddenly we get a new drug which never occurred in training data before." Yes, a new drug. But that drug (at least partly) relates to known ingredients, known symptoms, known side effects etc., i.e., to existing KB entities.

---

> > ### Author Response · Authors · 2021-11-27
> > **Further Response to Reviewer weC8**
> >
> > Thank you for the reply. Unfortunately, it seems many of the questions still come from a misunderstanding of the inductive learning setting and the problem; which we have spent a lot of text explaining.
> > > Q1. “W1 (convoluted). The authors argue that their approach uses prior algorithms in most of each steps. This does not make the approach less convoluted, however.”
> >
> > A1. First of all, we respectfully disagree that a method should be rejected just because it is “convoluted”. What we tried to explain is that although the method involves many steps, it is simply two stages: (1) mapping from input graph to cycle basis; (2) learning within the cycle space, and mapping the learning results back to triplets. As for why cycle space and cycle basis are necessary, please refer to our previous answers to **Q1 Reason to view rules as cycles; what if the good cycle is not in the basis** in the Overall Response. Given the motivation, all the steps are natural and clear algorithmic choices.
> >
> > An analogy can be drawn from the concept of neural networks, which is a folklore knowledge now, but indeed involves many steps, such as layers of nonlinear transformation, comparing with ground truth with loss functions, backpropagation, updating parameters accordingly, and optimization with stochastic gradient descent. The seemingly complex steps constitute a mature, widely-accepted, and efficient algorithm in practice. To some extent, our topology-based method is the same. Although it seems complicated at first, it is indeed a mature and widely-used method in computational topology.
> >
> > > Q2. “My concerns around learning erroneous rules ("related(x,z) and son(y, z) -> son(x,y)") have not been discussed.”
> >
> > A2. **Dealing with wrong rules.**
> > Indeed this question applies to all GNN methods and the answer is rather intuitive. Most of the learning methods can “learn” the confidence score of a rule. In detail, in our training stage, assume that there is a cycle in the SPT cycle bases corresponding to a wrong rule. There are two main possibilities that we can manage to learn the right rule / wrong rule:
> >
> > (1) There is a negative triplet (e.g., the target triplet is the negative triplet) in the cycle, the ground truth will be zero, therefore the model will assign a low confidence value to the rule after training.
> >
> > (2) All the triplets are positive triplets. Taking the example in Figure 1 (a), even if the chosen cycle basis contains wrong rule (e.g., the green cycle), we can learn the right blue cycle through the algebraic operation between the red and green cycles (or in practice, the message passing in the cycle graph).
> >
> > > Q3. “It's also not clear to me how all this relates to the statement "On the contrary, our method construct one unified GNN for all target triplets and learn/predict their confidence values simultaneously." made in the paper. How would one use such an approach in practice?”
> >
> > A3. In reality, there can be multiple target triplets that need to be predicted. Existing GNN-based inductive relation prediction methods need to extract the corresponding subgraph and predict them one by one (because the subgraphs are independent), while our methods can predict them simultaneously with higher speed. How can it be defined as not practical?
> >
> > > Q4. “((no analysis or theoretical insight). Expressivity or properties of the proposed method are neither discussed nor rigidly analyzed. Not addressed.”
> >
> > A4.**Expressivity analysis.** In terms of expressivity, our method is implicitly built on the following theorem: “Any cycle in the KG belongs to the cycle space, and thus can be expressed using a cycle basis, and thus be learned by our CBGNN.” We will make this explicit in the final version.
> >
> > **Theory insight.** Again, we have already addressed that “Our method is inspired by the theory of algebraic topology and computational topology (Munkres, 2018; Edelsbrunner & Harer, 2010; Dey & Wang, 2022)” and the whole method is based on such theory.

---

> > > ### Author Response · Authors · 2021-11-27
> > > **Addition Response to Reviewer weC8**
> > >
> > > > Q5. “The authors did not respond to my concerns around the asymptotic space/time complexity or reasons for the good empirical performance.”
> > >
> > > A5. We are surprised as we have explicitly addressed your concerns in **Evaluation of speed / computational efficiency**” in the Overall Response and **Q8 "there is no analysis or theoretical insight"**
> > >
> > > **Time complexity.** Again, we have already replied to you that “Please refer to the paragraph regarding “**Evaluation of speed / computational efficiency**” in the Overall Response.” In that part, we have discussed clearly why our method performs much better in terms of training and inference.
> > >
> > > **Reason for good empirical performance.** Again and again, we have already replied to you that “Any relevant cycle of the input KG is guaranteed to be represented by cycles in a cycle basis. In other words, we can represent and learn any relevant cycle, and thus any needed rule, with our CBGNN.” If a model can learn all the needed rules, then it can of course achieve a good performance.
> > >
> > > > Q6. “The example given in the response actually highlights the point made in my review, namely that entity overlap between train and test is likely. Yes, a new drug. But that drug (at least partly) relates to known ingredients, known symptoms, known side effects etc., i.e., to existing KB entities.”
> > >
> > > A6. As we have established, we are not the first paper to do inductive learning, and the practical value of this setting has been well realized. Inductive relation prediction tasks can be categorized into two types: semi-inductive learning and fully inductive learning (please refer to [1]). The two settings both have unseen entities in the test data, but for semi-inductive learning, the unseen entities are allowed to connect to seen entities, while in fully inductive learning, the test data is completely disconnected to the training data. The new drug case you mentioned is an example of semi-inductive learning, where the drug has connections to the existing entities.
> > >
> > > In our paper, we follow the fully inductive setting of GRAIL[2], and test entities do not have connections to the training entities. It is common that many KGs are separated in the real-world. Although the information of drugs can often be shared, in many cases the drugs/diseases in different KGs do not have identical IDs and these KGs are still separated. A more intuitive example for fully inductive learning may be the event graphs for different news clusters (in most cases, these events do not have the same named entities as nodes).
> > >
> > > [1] Ali et al. Improving Inductive Link Prediction Using Hyper-Relational Facts. In ISWC 2021.
> > >
> > > [2] Teru et al. Inductive Relation Prediction by Subgraph Reasoning. In ICML2020.

---

### Official Review · Reviewer_vHmE · 2021-11-02

**Correctness:** 2
**Technical Novelty And Significance:** 2
**Empirical Novelty And Significance:** 2
**Recommendation:** 5
**Confidence:** 4

**Main Review:**

Overall the paper is well-written and easy to follow. I find the idea of viewing relational paths as graph cycles to be interesting. However, despite the authors claiming that cycle representation is distinct from the path representation in multi-hop reasoning methods such as NeuralLP, it's worth noting that the path representation is guaranteed to be a cycle in these works (as a chain rule). Therefore, the search space of the two representations is in fact the same. I would suggest the author carefully address the claims and comparisons with the multi-hop reasoning methods in sections 1 and 2.


Some of the concepts and design choices need more elaborations and justifications:
 - The proposed method generates nodes with the spectral clustering algorithm. It's unclear why this particular algorithm is picked and how sensitive is the framework to the choice of clustering algorithms.
 - The authors generate cycles with SPT. However, other than claiming it follows Occam’s razor principle, there is no theoretical analysis or direct empirical experiment that shows the space always contains the answer cycle to the query.
 - The authors claim the cycle incidence matrix implicitly encodes interaction between the cycles. However, it's unclear why such interaction information is needed for answering the query. To justify this, the author should compare the method with a simpler baseline without the incidence matrix.

Cycle extraction only considers the KG's topological structure, treating the KG as an undirected graph with no edge label. I'm concerned about this approach, as the direction and the type of edge are critical to the KG. In fact, in the feature generation phase, the edge direction and type are indeed taken into consideration. It's unclear to me why not use this information at the very beginning.

Besides the aforementioned issues, my main concern with this work is that the proposed framework is unnecessarily complicated.
 - It generates cycles with clustering and SPT -> computes cycle incidence matrix -> generates cycle features with bi-LSTM -> builds GNN for each cycle basis -> computes cycle confidence with GNN -> aggregate final scores with weighted sum.
 - The current ablation study is far from sufficient to justify why this framework has to be this complicated.
 - For example, one can build a frequency-based baseline with cycles generated from SPT and compare it with the proposed method.
 - For the cycle incidence matrix, one can build a baseline without the matrix.
 - To predict the edges, one can also build a simpler classifier right on top of the bi-LSTM model. It's critical to show that the additional GNN structure is indeed necessary.

I'm also concerned about the method scalability. The computational costs of the proposed method grow quickly w.r.t |V|, |E| and beta. However, the largest dataset FB15K v5 only contains 5K nodes and 33K edges which is significantly smaller than the original FB15K dataset which contains 14K nodes and 272K edges.
 - It's unclear to me what does the run time shown in Table 3 correspond to. Does the time correspond to the end-to-end processing time of CBGNN? Or it is the inference time of the GNN module alone?

The analysis on the influence of k is weak. The experiment is only conducted on the smallest subset of FB15K and WN18. Instead, the authors should analyze the necessary k values against a set of KGs of increasing sizes and show how well does the k scales w.r.t |V|, |E| and beta.

**Summary Of The Paper:**

The paper proposes a GNN framework to solve the inductive relation prediction problem, namely CBGNN. The framework consists of three modules: 1)  a cycle extraction module that generates cycles graphs with clustering and SPT tree search; 2) a bi-directional LSTM that converts a cycle of triplets into a feature vector, and 3) a GNN model that learns the triplet confidence using the feature vectors. In experiments, the proposed method is evaluated with a set of differentiable ILP methods and GNN methods. Some ablation studies are conducted to validate the cycle extraction module and feature generation module.


**Summary Of The Review:**

In summary, although the idea of using cycle representation is interesting. The proposed method is unnecessarily complicated. And the current draft lacks too many justifications to the components in the proposed method. At this point, I would recommend rejection but I'm happy to raise my score if the authors address the aforementioned concerns.

---

> ### Author Response · Authors · 2021-11-23
> **Response to Reviewer vHmE**
>
> > Q1. “the search space of the two representations is in fact the same. I would suggest the author carefully address the claims and comparisons with the multi-hop reasoning methods in sections 1 and 2.”
>
> A1. Please refer to the paragraph regarding **“Q1 Reason to view rules as cycles”** in the Overall Response.
>
> > Q2. “The proposed method generates nodes with the spectral clustering algorithm. It's unclear why this particular algorithm is picked and how sensitive is the framework to the choice of clustering algorithms.”
>
> A2. **Reason to choose spectral clustering.**
> We are mainly looking for a sampling of nodes that can spread out evenly over the graph. A convenient choice was spectral clustering, which was used to select anchor nodes in other contexts. We are open to any other suggestions.
>
> > Q3. “The authors generate cycles with SPT. However, other than claiming it follows Occam’s razor principle, there is no theoretical analysis or direct empirical experiment that shows the space always contains the answer cycle to the query.”
>
> A3. **Cycle space always contains the answer cycle (rules) .**
> The cycle space represented by a cycle basis is guaranteed to contain all possible cycles in the input graph. We are using the SPT cycle bases to represent the needed rules rather than directly choosing from the cycles in the SPT cycle bases. Please refer to the paragraph regarding **“Q1 Reason to view rules as cycles”** in the Overall Response for detailed discussion. As for the shortness of cycles, please refer to the paragraph regarding **“Q2 Length of good cycles”** in the Overall Response.
>
> > Q4. “The authors claim the cycle incidence matrix implicitly encodes interaction between the cycles. However, it's unclear why such interaction information is needed for answering the query. To justify this, the author should compare the method with a simpler baseline without the incidence matrix.”
>
> A4. **The use of cycle incidence matrix.**
> The cycle incidence matrix is needed to map the weights of good cycles down to triplets for the final prediction. Meanwhile, it is used to measure overlapping/interaction between cycles in a basis. This cycle interaction graph, as well as the message passing over it, is essential to our central hypothesis: through message-passing between cycles of a basis, we can find the relevant cycles in the cycle space. A relevant cycle is not necessarily a cycle in the basis, but is represented as a sum of cycles in the basis, and can be learned by the Cycle-GNN. A related discussion is in Q1 of the Overall Response.
>
> To prove the necessity of learning the needed rules rather than directly choosing cycles from the cycle bases, we add a new baseline CBGNN-MLP in Table 4. CBGNN -noGNN does not use a GNN model to perform the operation between cycles in the cycle bases, but directly uses a BRLSTM followed by a two-layer MLP to get the confidence of cycles. As is shown in Table 4, CBGNN-MLP consistently performs worse than CBGNN, showing the necessity of learning the needed rules with the cycle bases. In addition, the result of CBGNN-MLP is comparable with state-of-the-art inductive relation prediction methods, showing that the SPT cycle bases contain a number of right cycles (the needed rules). This provides empirical observation that SPT cycle bases are generally “suitable cycle bases".

---

> > ### Author Response · Authors · 2021-11-23
> > **Additional Response to Reviewer vHmE**
> >
> > > Q5. “Cycle extraction only considers the KG's topological structure, treating the KG as an undirected graph with no edge label. I'm concerned about this approach, as the direction and the type of edge are critical to the KG. In fact, in the feature generation phase, the edge direction and type are indeed taken into consideration. It's unclear to me why not use this information at the very beginning.”
> >
> > A5. **Extracting undirected cycles in SPT cycle bases.**
> > Inspired by existing methods on rule learning (Marcheggiani & Titov, 2017; Vashishth et al., 2019), we assume that in the knowledge graph, the information in triplet flows along both directions, and has empirically proven the assumption in Table 4 (CBGNN and CBGNN-LSTM). Therefore, all the rules can be represented by two cycles with different directions (an illustration can be found in Figure 3 in the appendix). In our case, we use undirected cycles in the SPT cycle bases and BRLSTM to discriminate the two cycles with different directions.
> >
> > We note that directly modeling cycles of a directed graph is a challenging problem and an active research direction in computational topology [1-4]. We work with the undirected version of KG and bi-directional modeling of cycles (BRLSTM) to circumvent the computational and modeling challenges. This is an interesting future direction to explore.
> >
> >  [1] Chowdhury and Mémoli. "Persistent path homology of directed networks." In Proceedings of the Twenty-Ninth Annual ACM-SIAM Symposium on Discrete Algorithms (SODA), 2018.
> >
> > [2] Chowdhury and Mémoli. "A functorial Dowker theorem and persistent homology of asymmetric networks." Journal of Applied and Computational Topology 2, no. 1 (2018): 115-175.
> >
> > [3] Grigor'yan, Alexander, Rolando Jimenez, Yuri Muranov, and Shing-Tung Yau. "Homology of path complexes and hypergraphs." Topology and its Applications 267 (2019): 106877.
> >
> > [4] Dey, Tamal K., Tianqi Li, and Yusu Wang. "An Efficient Algorithm for 1-Dimensional (Persistent) Path Homology." In 36th International Symposium on Computational Geometry (SoCG), 2020.
> >
> > > Q6.” Besides the aforementioned issues, my main concern with this work is that the proposed framework is unnecessarily complicated.”
> >
> > A6. Please refer to the paragraph regarding **“Q3 The complicatedness and computational efficiency of our model”** in the Overall Response.
> >
> > > Q7. "I'm also concerned about the method scalability. "
> >
> > A7. **Scalability.**
> > Actually, we are more scalable compared with existing inductive relation prediction methods, please refer to the paragraph regarding **“Evaluation of speed / computational efficiency”** in the Overall Response.
> >
> > > Q8. "The analysis on the influence of k is weak. "
> >
> > A8. **influence of k.**
> > To evaluate the influence of k more thoroughly, we add extra experiments on 2 larger datasets WN18RR v1 and FB-15k 237 v2, the results are shown in Figure 4 in the appendix.

---

> > > ### Comment · Reviewer_vHmE · 2021-11-29
> > > **Thoughts after reading the rebuttal**
> > >
> > > Thank you for the rebuttal and the revised draft. The new experiments definitely go in the right direction. My concerns on the necessity of each proposed module and model scalability are partially addressed. While I understand it's difficult to conduct all ablation studies during the response phase, the current draft still lacks sufficient theoretical and empirical support to justify the proposed method and its complexity. For example, 1) selection of clustering algorithms; 2) frequency-based baseline showing how much does the cycle representation contribute to the performance vs. the learning part built on top of it; 3) asymptotic analysis of k w.r.t the size of the graph (figure 4 is still lacking); and so on. Presumably, with all expected changes in place, it would significantly alter the draft and will require a new round of reviews to ensure everything is in place. That said, I'm keeping my score at 5.

---

### Official Review · Reviewer_YAZM · 2021-11-03

**Correctness:** 2
**Technical Novelty And Significance:** 3
**Empirical Novelty And Significance:** 3
**Recommendation:** 3
**Confidence:** 4

**Main Review:**

strengths:
* The problem of learning rules for inductive relation prediction is important.
* The authors provided another view, circle-based rules, about learning logical rules.
* The authors conduct experiments on benchmarks and the method achieves good performance.

weaknesses:
* It’s not clear what can be done by the proposed circle-based rules but cannot be done by previous chain-based rules. Actually, all circle-based rules can be regarded by chain-like rules, and by optimizing the scores of chain-like rules, one can learn high-quality rules.
* Suppose we have learned a circle (A, B, C), if we want to rewrite it as a rule, which of the following should we use?
A, B->C; A, C->B; B, C->A.
In my opinion, they are significantly different, but it seems that the proposed method cannot distinguish them based on the circle-based methods.
* The training time and the testing time should be reported separately.
* It would be better to use early-stopping rather than training with a fixed number of epochs for performance comparison.
* “We hypothesize that the good cycles tend to be short, and the desired cycle bases should generally contain short cycles.”
Personally, I don’t think it is convincing to me, because some rules are long but interesting. Actually, according to previous literature[1,2,3], the rules with lengths of 1,2,3 are all important. The examples are as follows,
film_edited_by(B,A)←nominated_for(A,B);
 partially_contains(C,A)←contains(B,A) ∧ contains(B,C);
Person(X) ← Car(Y1) ∧ Inside(X, Y1) ∧On(Y2, X) ∧ Clothing(Y2).
* It would be better if the authors also compare their method with NLIL[3], which can learn more complex rules.
* Minor issue (typo):
O(|E||V|).


[1] Yang, Fan, Zhilin Yang, and William W. Cohen. "Differentiable learning of logical rules for knowledge base reasoning." Proceedings of the 31st International Conference on Neural Information Processing Systems. 2017.

[2] Sadeghian, Ali, et al. "DRUM: End-To-End Differentiable Rule Mining On Knowledge Graphs." Advances in Neural Information Processing Systems 32 (2019): 15347-15357.

[3] Yang, Yuan, and Le Song. "Learn to Explain Efficiently via Neural Logic Inductive Learning." International Conference on Learning Representations. 2019.



**Summary Of The Paper:**

This paper focuses on the problem of inductive relational prediction for KG completion. Previous works often regard the rules as paths, different from those works, this paper regards the rules as circles. Based on that, the authors propose a GCN-based method for learning rules. They conduct experiments on the public datasets of inductive relation prediction and the proposed method achieves better or similar performance compared to best baselines.

**Summary Of The Review:**

The paper proposed a circle-based method for rule learning, which is an interesting view. However, some claims (e.g., shorter rules are better) are not convincing, and the motivation of such a design is not clear. There is no substantial difference between the rule learned by this method and the previous works, and the search space of rules is even smaller (e.g., compared to NLIL[3]). I recommend the authors clarify these issues.

---

> ### Author Response · Authors · 2021-11-23
> **Response to Reviewer YAZM**
>
> > Q1. “It’s not clear what can be done by the proposed circle-based rules but cannot be done by previous chain-based rules. Actually, all circle-based rules can be regarded by chain-like rules, and by optimizing the scores of chain-like rules, one can learn high-quality rules.”
>
> A1. Please refer to the paragraph regarding **“Q1 Reason to view rules as cycles”** in the Overall Response.
>
>
>
> > Q2. “Suppose we have learned a circle (A, B, C) if we want to rewrite it as a rule, which of the following should we use? A, B->C; A, C->B; B, C->A. In my opinion, they are significantly different, but it seems that the proposed method cannot distinguish them based on the circle-based methods.”
>
> A2. **The direction/sequence of rules.**
> In our settings, for every cycle/rule, we only use one sequence to represent it, that is, to set the target triplet (the non-tree edge in practice, as stated in Section 4.1) as the start token of BRLSTM. Our aim is to predict the target triplet through the other triplets in the cycle. For example, in Figure 3 in the appendix, if we want to predict the existence of the triplet “(Cristiano,  part of, ManchesterUnited)”, we set it as the first token in the BRLSTM, as explained in Section A.2.
>
> > Q3. “The training time and the testing time should be reported separately.”
>
> Please refer to the paragraph regarding **“Evaluation of speed / computational efficiency”** in the Overall Response.
>
> > Q4. “It would be better to use early-stopping rather than training with a fixed number of epochs for performance comparison.”
>
> A4. **Early-stopping.**
> In the training stage, we set early-stopping as 20. In the evaluation of computational efficiency, we remove early-stopping to make all the training epochs the same.
>
> > Q5.  “We hypothesize that the good cycles tend to be short, and the desired cycle bases should generally contain short cycles.” Personally, I don’t think it is convincing to me, because some rules are long but interesting. Actually, according to previous literature[1,2,3], the rules with lengths of 1,2,3 are all important. The examples are as follows, film_edited_by(B,A)←nominated_for(A,B); partially_contains(C,A)←contains(B,A) ∧ contains(B,C); Person(X) ← Car(Y1) ∧ Inside(X, Y1) ∧On(Y2, X) ∧ Clothing(Y2).”
>
> A5. Please refer to the paragraph regarding **“Q2 Length of good cycles”** in the Overall Response.
>
> > Q6. "It would be better if the authors also compare their method with NLIL, which can learn more complex rules."
>
> A6. **Comparison with NLIL.** Actually, the setting of NLIL [1] is quite different from the setting of inductive relation prediction.
>
> (1) In the setting of NLIL, existing triplets are classified into “training/validation/test sets” and “fact set”. The former sets are for model training and inference and do not provide information to other triplets in these sets, while the latter set provides the information of knowledge graphs to the former sets. However, in the setting of inductive relation prediction (Teru et al., 2020; Mai et al., 2021), existing triplets are classified into “training/validation/test sets”, which also serve as the “fact set” in the training and inference stage. As for the negative (originally non-existent) triplets, they are temporarily added to the KG in the generation of cycle bases / the extraction of subgraphs.
>
> (2) In the setting of NLIL, the test set and the training set can have overlapped entities, while in the setting of inductive relation prediction, they can not.
> Therefore, when using the default setting of WN18 ( and FB15k-237 resp.) in NLIL to evaluate WN18RR v1 (FB15k-237 v1 resp.), we find that the Hits@10 scores are very low in WN18RR v1 and v4, and for FB15k-237 v1, even if setting the batch size to 1 will lead to a “CUDA out of memory” error for RTX 2080TI. We are happy to improve the results according to the reviewers’ suggestions.
>
>
> |Dataset |WN18RR v1 | WN18RR v4|
> |:------:|:------:|:------:|
> |MRR|0.0014|0.0002|
> |Hits@10|0.0000|0.0000|
>
>
> [1] Yang Y, Song L. Learn to Explain Efficiently via Neural Logic Inductive Learning. ICLR 2019.

---

### Author Response · Authors · 2021-11-23
**Overall Response**

Based on the reviewers’ request, we have revised the paper. Changes are highlighted in blue. The major revisions are listed as follows:

**(1) Necessity of the GNN and a new baseline.** Per reviewers vHmE and weC8’s request, we added a new baseline CBGNN-MLP in order to validate the necessity of GNNs on the cycle graphs. CBGNN-MLP is an ablation model which removes the GNN component and directly chooses the needed rules from the cycles in the cycle bases. Its performance is provided below (and in Table 4). CBGNN-MLP consistently performs worse than our method, showing that the needed rules are not always in the SPT cycle bases. Therefore, we need to use GNNs to learn the right rules by exploring the whole space of exponentially many cycles, not directly choosing from cycles of the SPT bases. See Q1 below for more related discussions.

| Datasets | |WN18RR| | | | FB15k-237| | | | NELL-995 |||
|:------:|:------:|:------:|:------:|:------:|:------:|:------:|:------:|:------:|:------:|:------:|:------:|:------:|
|Split |v1|v2|v3|v4|v1|v2|v3|v4|v1|v2|v3|v4|
|CBGNN-MLP| 96.33  | 97.49  | 86.86 | 95.22 | 90.90 | 94.07 | 87.01 | 87.78  | 72.29 | 93.35 | 94.63 | 91.29 |
| CBGNN | **98.63** | **97.62** | **89.76** | **97.80** | **96.34** | **96.53**| **96.38**|**95.23** | **82.79** | **78.79** |**94.78** | **96.29** | **94.02** |


**(2) Evaluation of speed / computational efficiency.** We updated the evaluation on the efficiency of the models in Table 3 (and below) by separating the preparation, training, and inference time. As shown in the table below, the generation of SPT cycle bases costs no more than the extraction of subgraphs, while our model is much superior to the baseline methods with regard to the training time and inference time. This results from the complicated GNN learning strategy in Grail (Teru et al., 2020) and CoMPILE (Mai et al., 2021), which needs to train the GNN model on all the subgraphs, while our model only needs to be trained on the cycle graph.


| Dataset| |WN18RR|  | | FB15k-237| |  | NELL-995 ||
|:------:|:------:|:------:|:------:|:------:|:------:|:------:|:------:|:------:|:------:|
| Phase | Preparation | Training | Inference | Preparation | Training | Inference |  Preparation | Training | Inference |
| GraIL | 452.36 | 2230.55 | 1.07 | 704.42 | 9026.21 | 1.67   |  402.86 | 3718.22 |  1.79 |
|CoMPILE | **434.45** | 2388.28 | 1.46 | 706.19 | 3809.56 | 2.41 | 479.21 | 2868.38 | 1.23|
|CBGNN | 601.96 | **952.55** | **0.52** | **437.13** | **901.27** | **0.75** | **379.29** | **175.19** | **0.14**|

**(3) Influence of k.** We further evaluate the influence of k (the number of SPT cycle bases) on WN18RR v2 and FB15k-237 v2 in Section A.3 the appendix.

---

> ### Author Response · Authors · 2021-11-23
> **Additional Overall Response**
>
> There are some common concerns from the reviewers and we provide the answer below to save the space and also make it more clear.
>
> **Q1 Reason to view rules as cycles; what if the good cycle is not in the basis**
>
> We are not restricting the learning to only cycles in a basis as candidates. The key benefit of a cycle-centric approach is that we can represent an exponential-size cycle space with just a linear-size cycle basis. A cycle basis with n cycles can represent $2^n-1$ possible cycles; any cycle can be uniquely written as a linear sum of the basis cycles with 0/1 coefficients. Our hypothesis is that all relevant cycles in the cycle space, whether positively or negatively relevant, can be represented with and be learned using the cycle basis. The GNN is essentially operating on the cycles of the basis and imitating the algebraic operations. We will clarify this in the paper.
> On the contrary, path-based methods attempt to explicitly capture all relevant paths. They have to approximate the weight of a whole path with the product of the weights of relations in the path. This approximation is rather coarse and results in rather unsatisfying performance, as shown in Tables 1 and 2. A detailed discussion can be found in our Related Works section.
>
>
> **Q2 Length of good cycles**
>
> First, we would like to clarify the disambiguation between “the short cycles in the cycle bases” and “the right rules”/”the needed rules”. The cycles in the cycle bases are not the cycle rules that are directly used for relation prediction; instead, our aim is to learn the needed rules with these cycles in the cycle bases. Therefore, although the bases generally contain short cycles, the learned rules can be long. For example, in Figure 1 (a), the length-4 green cycle can be represented by the two length-3 red and blue cycles. In practice, short cycles can be learned more efficiently, therefore we assume that cycles in the cycle bases should generally be short. We will clarify this part in the paper.
> In addition, SPT cycle bases do not only contain short cycles. It is shown in Figure 5 in the appendix that the SPT cycle bases contain cycles longer than 10, and the proportion of these cycles is larger than 20%. On the contrary, existing works (Yang et al., 2017; Sadeghian et al., 2019; Teru et al., 2020; Mai et al., 2021) limit the length of learned rules to a small number in case of high computational cost, and cannot represent long rules in real settings.
>
> **Q3 The complicatedness and computational efficiency of our model**
>
> **Computational efficiency.** Compared with existing works on inductive relation prediction, our framework strikes a good balance between effectiveness and computational efficiency.
> Path-based methods (Yang et al., 2017; Sadeghian et al., 2019) generally perform unsatisfyingly. Neural LP (Yang et al., 2017) will bring in wrong rules, and DRUM (Sadeghian et al., 2019) needs a large number of learnable parameters to correctly learn all the rules. In reality, DRUM only uses a limited-size model to learn rules, thus their performance is not satisfying, as shown in Tables 1 and 2.
> GNN-based methods (Teru et al., 2020; Mai et al., 2021) are much slower than our framework, besides, their performances are also worse than ours. For every triplet, these models need to extract the corresponding subgraph and use a GNN to learn, therefore they can only predict the triplet one by one. On the contrary, our methods can predict all the target triplets in a single learning process on the whole graph. The comparison of efficiency is shown in Table 3.
>
> **Complicatedness.** As explained in our answer to Q1, we want to explore the exponential-size space of all cycles to learn the relevant cycles, but without blowing up the number of parameters. This is built on the insight that the cycle space is represented by the cycle basis. The seemingly complex steps essentially boils down to two stages: (1) mapping from input graph to cycle basis; (2) learning within the cycle space, and mapping the learning results back to triplets. The first part is carried out by classic algorithms from computational topology. The second part is carried out through a GNN passing messages between basis cycles. This is a good example of how theoretical insights can help us simplify the complex task of learning.

---

### Decision · Program_Chairs · 2022-01-20

**Decision:**

Reject

**Comment:**

The paper proposes a new approach to inductive rule prediction for knowledge graph completion. Reviewers highlighted as strengths that the paper proposes an interesting approach to an important problem that is relevant for the ICLR community. However, reviewers raised also concerns regarding model design and correctness as well as clarity of presentation (e.g., motivation, analysis, comparison to related work, evaluation). After author response and discussion, all reviewers and the AC agree that the paper is not yet ready for publication at ICLR due to the aforementioned issues.